# N-terminal domain on dystroglycan enables LARGE1 to extend matriglycan on α-dystroglycan and prevents muscular dystrophy

Hidehiko Okuma[1], Jeffrey M Hord[1], Ishita Chandel[1], David Venzke[1], Mary E Anderson[1], Ameya S Walimbe[1], Soumya Joseph[1], Zeita Gastel[1], Yuji Hara[2], Fumiaki Saito[3], Kiichiro Matsumura[3], Kevin P Campbell[1]*

[1]Howard Hughes Medical Institute, Senator Paul D. Wellstone Muscular Dystrophy Specialized Research Center, Department of Molecular Physiology and Biophysics and Department of Neurology, Roy J. and Lucille A. Carver College of Medicine, The University of Iowa, Iowa City, United States; [2]Department Pharmaceutical Sciences, School of Pharmaceutical Sciences, University of Shizuoka, Shizuoka, Japan; [3]Department of Neurology, School of Medicine, Teikyo University, Tokyo, Japan

*For correspondence:
kevin-campbell@uiowa.edu

Competing interest: The authors declare that no competing interests exist.

**Abstract** Dystroglycan (DG) requires extensive post-translational processing and O-glycosylation to function as a receptor for extracellular matrix (ECM) proteins containing laminin-G (LG) domains. Matriglycan is an elongated polysaccharide of alternating xylose (Xyl) and glucuronic acid (GlcA) that binds with high affinity to ECM proteins with LG domains and is uniquely synthesized on α-dystroglycan (α-DG) by like-acetylglucosaminyltransferase-1 (LARGE1). Defects in the post-translational processing or O-glycosylation of α-DG that result in a shorter form of matriglycan reduce the size of α-DG and decrease laminin binding, leading to various forms of muscular dystrophy. Previously, we demonstrated that protein O-mannose kinase (POMK) is required for LARGE1 to generate full-length matriglycan on α-DG (~150–250 kDa) (Walimbe et al., 2020). Here, we show that LARGE1 can only synthesize a short, non-elongated form of matriglycan in mouse skeletal muscle that lacks the DG N-terminus (α-DGN), resulting in an ~100–125 kDa α-DG. This smaller form of α-DG binds laminin and maintains specific force but does not prevent muscle pathophysiology, including reduced force production after eccentric contractions (ECs) or abnormalities in the neuromuscular junctions. Collectively, our study demonstrates that α-DGN, like POMK, is required for LARGE1 to extend matriglycan to its full mature length on α-DG and thus prevent muscle pathophysiology.

## Editor's evaluation

This study presents fundamental new insight into the process of post-translational modification of α dystroglycan by the protein Large on its N-terminal domain which is critical for concentric muscle contraction. The convincing data presented advances the field beyond a role for POMK in mediating the effect of Large on α dystroglycan, to show that α dystroglycan N-term domain, like POMK itself, is required for LARGE1 to extend matriglycan to its full mature length.

## Introduction

The basement membrane is a specialized network of extracellular matrix (ECM) macromolecules that surrounds epithelium, endothelium, muscle, fat, and neurons (*Rowe and Weiss, 2008*). Skeletal muscle

cells are bound to the basement membrane through transmembrane receptors, including dystroglycan (DG) and integrins, which help maintain the structural and functional integrity of the muscle cell membrane (*Ibraghimov-Beskrovnaya et al., 1992*; *Han et al., 2009*). DG is a central component of the dystrophin-glycoprotein complex (DGC). It is encoded by a single gene, *DAG1*, and cleaved into α- and β-subunits (α-DG and β-DG, respectively) by post-translational processing (*Ibraghimov-Beskrovnaya et al., 1992*). Extensive *O*-glycosylation of α-dystroglycan (α-DG) is required for normal muscle function, and defects in this process result in various forms of muscular dystrophy (*Michele et al., 2002*; *Ohtsubo and Marth, 2006*; *Yoshida-Moriguchi and Campbell, 2015*; *Praissman et al., 2016*; *Sheikh et al., 2017*).

α-DG binds to ECM ligands containing laminin-G (LG) domains (e.g., laminin, agrin, perlecan) which are essential components of the basement membrane (*Michele et al., 2002*). DG, thereby, physically links the cell membrane to the basement membrane. This process requires the synthesis of matriglycan, a heteropolysaccharide [-GlcA-β1,3-Xyl-α1,3-]$_n$, on α-DG by the bifunctional glycosyltransferase, like-acetylglucosaminyltransferase-1 (LARGE1) (*Inamori et al., 2012*; *Yoshida-Moriguchi and Campbell, 2015*; *Hohenester, 2019*; *Ohtsubo and Marth, 2006*). Structural studies have shown that a single glucuronic acid-xylose disaccharide (GlcA-Xyl) repeat binds with high-affinity to laminin-α2 LG4,5, and there is a direct correlation between the number of GlcA-Xyl repeats on α-DG and its binding capacity for ECM ligands (*Briggs et al., 2016*). During skeletal muscle differentiation, LARGE1 elongates matriglycan, thereby increasing the binding capacity of matriglycan for ECM ligands and allowing matriglycan to serve as a matrix scaffold that is required for skeletal muscle function (*Goddeeris et al., 2013*). Matriglycan synthesis requires the addition of core M3 trisaccharide (GalNAc-β1,3-GlcNAc-β1,4-Man) onto specific threonine residues on α-DG in the endoplasmic reticulum (ER) (*Sheikh et al., 2017*). Once core M3 is synthesized, the glycosylation-specific kinase, protein *O*-mannose kinase (POMK) phosphorylates mannose of core M3, and this is required for the subsequent addition of full-length, high-molecular weight forms of matriglycan onto the *O*-mannose linked modification (*Yoshida-Moriguchi and Campbell, 2015*; *Hohenester, 2019*; *Jae et al., 2013*; *Yoshida-Moriguchi et al., 2013*; *Zhu et al., 2016*). In the absence of core M3 phosphorylation by POMK, LARGE1 synthesizes a short, non-elongated form of matriglycan on α-DG (*Walimbe et al., 2020*). Notably, a loss of function in the post-translational addition of matriglycan leads to a group of disorders known as dystroglycanopathies, congenital and limb-girdle muscular dystrophies with

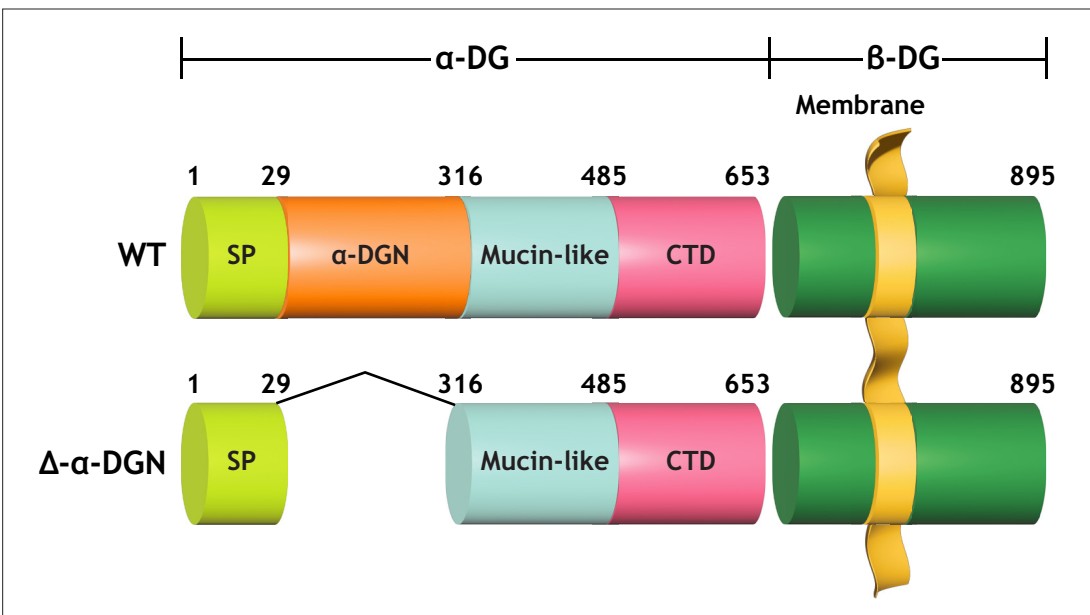

**Figure 1.** Domain structure of dystroglycan (DG) and Δ-α-DGN. Wild-type DG is a pre-proprotein with an N-terminal signal peptide (light green) that is translated in the rough endoplasmic reticulum. The globular N-terminal domain (α-DGN; orange) is present in wild-type DG but absent in the mutant (Δ-α-DGN). The junction between α-DGN and the mucin-like domain (light teal) contains a furin convertase site. The globular extracellular C-terminal domain (CTD; pink) contains an SEA (sea urchin sperm protein, enterokinase and agrin) autoproteolysis site, which cleaves pro-DG into α-DG and β-DG (green). Glycosylation has been omitted for clarity.

or without brain and eye abnormalities. Disease severity is dependent on the ability of matriglycan to bind ECM ligands, which is dictated by its length and expression (*Goddeeris et al., 2013*): matriglycan that is of low molecular weight (e.g., short) can cause muscular dystrophy, even if its capacity to bind LG domains is not completely lost (*Puckett et al., 2009*; *Hara et al., 2011a*; *Hara et al., 2011b*; *Carss et al., 2013*; *Cirak et al., 2013*; *Dong et al., 2015*; *Walimbe et al., 2020*). However, how matriglycan elongation is regulated by factors other than POMK is still unknown.

α-DG is composed of three distinct domains: the N-terminal (α-DGN) domain, a central mucin-like domain, and the C-terminal domain (*Figure 1*; *Ibraghimov-Beskrovnaya et al., 1992*; *Brancaccio et al., 1995*; *Brancaccio et al., 1997*). α-DGN serves as a binding site for LARGE1 in the Golgi and is required for the subsequent functional glycosylation of the mucin-like domain of α-DG (*Kanagawa et al., 2004*). We, therefore, hypothesized that α-DGN must be involved in regulating the production and elongation of matriglycan. Here, we have used a multidisciplinary approach to show that LARGE1 synthesizes a non-elongated form of matriglycan on DG that lacks α-DGN (i.e., α-DGN-deleted DG) resulting in ~100–125 kDa α-DG. This short form of matriglycan binds laminin and maintains muscle-specific force. However, it fails to prevent lengthening contraction-induced reductions in force, neuromuscular junction (NMJ) abnormalities, or dystrophic changes in muscle. Collectively, our study shows that LARGE1 requires α-DGN to generate full-length matriglycan in skeletal muscle, but the synthesis of a shorter form of matriglycan can proceed independently of this domain.

## Results

Constitutive deletion of DG in mice causes embryonic lethality due to disruption of Reichert's membrane, an extraembryonic basement membrane required for in utero development (*Williamson et al., 1997*). Deletion of α-DGN in mice also causes embryonic lethality (*de Greef et al., 2019*). However, mice that are heterozygous for α-DGN deletion (*Dag1^{wt/Δα-DGN}*) are viable and express α-DG of two different sizes (*Figure 2—figure supplement 1*) corresponding to both wild-type (WT) and the α-DGN-deleted (Δα-DGN) forms of DG. Thus, to successfully ablate α-DGN in skeletal muscle, we used mice expressing Cre recombinase under the control of the *paired Box 7* (*Pax7*) promoter (*Pax7^{Cre}*), floxed DG mice (*Dag1^{flox/flox}*), and heterozygous α-DGN-deleted mice (*Dag1^{wt/Δα-DGN}*) to generate *Pax7^{Cre}*; *Dag1^{flox/Δα-DGN}* (muscle-specific α-DGN knockout [M-α-DGN KO]) mice (*Figures 1 and 2*).

To evaluate the gross phenotype of mice harboring a muscle-specific deletion of α-DGN (i.e., M-α-DGN KO mice), we first measured body weight and grip strength. M-α-DGN KO mice were slightly lower in weight than WT littermate (control) mice at 12 weeks of age, and they exhibited decreased forelimb grip strength as well (*Figure 2A*). To determine whether deletion of α-DGN affects matriglycan expression, we performed histological analysis of quadriceps muscle from control and M-α-DGN KO mice. M-α-DGN KO mice showed characteristic features of muscular dystrophy, including an increase in centrally nucleated fibers (*Figure 2B*). Immunofluorescence analyses of M-α-DGN KO muscle showed reduced levels of matriglycan relative to controls, but a similar expression of β-DG, the transmembrane subunit of DG (*Figure 2B*).

Immunoblot analysis demonstrated that skeletal muscle from M-α-DGN KO mice expresses a shorter form of matriglycan which results in an ~100–125 kDa α-DG, a decrease in the molecular weight of the core α-DG, and no change in β-DG (*Figure 2C*). No matriglycan is seen in *Large^{myd}* mice which have a deletion in *Large1* (*Figure 2C*). To investigate how the loss of α-DGN affects ligand binding, we performed a laminin overlay using laminin-111. Skeletal muscle from control mice showed a broad band centered at ~100–250 kDa, indicative of α-DG-laminin binding; in contrast, we observed laminin binding at ~100–125 kDa in M-α-DGN KO skeletal muscle (*Figure 2C*). To further confirm that the ~100–125 kDa band seen with anti-matriglycan antibodies in M-α-DGN KO muscle is matriglycan-positive α-DG, we digested it overnight with β-glucuronidase (*Thermotoga maritima*) and α-xylosidase (*Sulfolobus solfataricus*). Immunoblot analysis with anti-matriglycan antibodies and laminin overlay revealed that ~100–125 kDa was completely lost after enzymatic digestion, indicating that the ~100–125 kDa band is indeed matriglycan-positive α-DG (*Figure 2—figure supplement 2*). The sensitivity of ~100–125 kDa α-DG to enzymatic digestion also demonstrates that matriglycan is not capped like brain α-DG (*Sheikh et al., 2020*).

The NMJs in adult control mice demonstrate a normal pretzel-like morphology whereas NMJs from M-α-DGN KO mice display a variety of abnormalities, including a granular appearance and AChR-rich

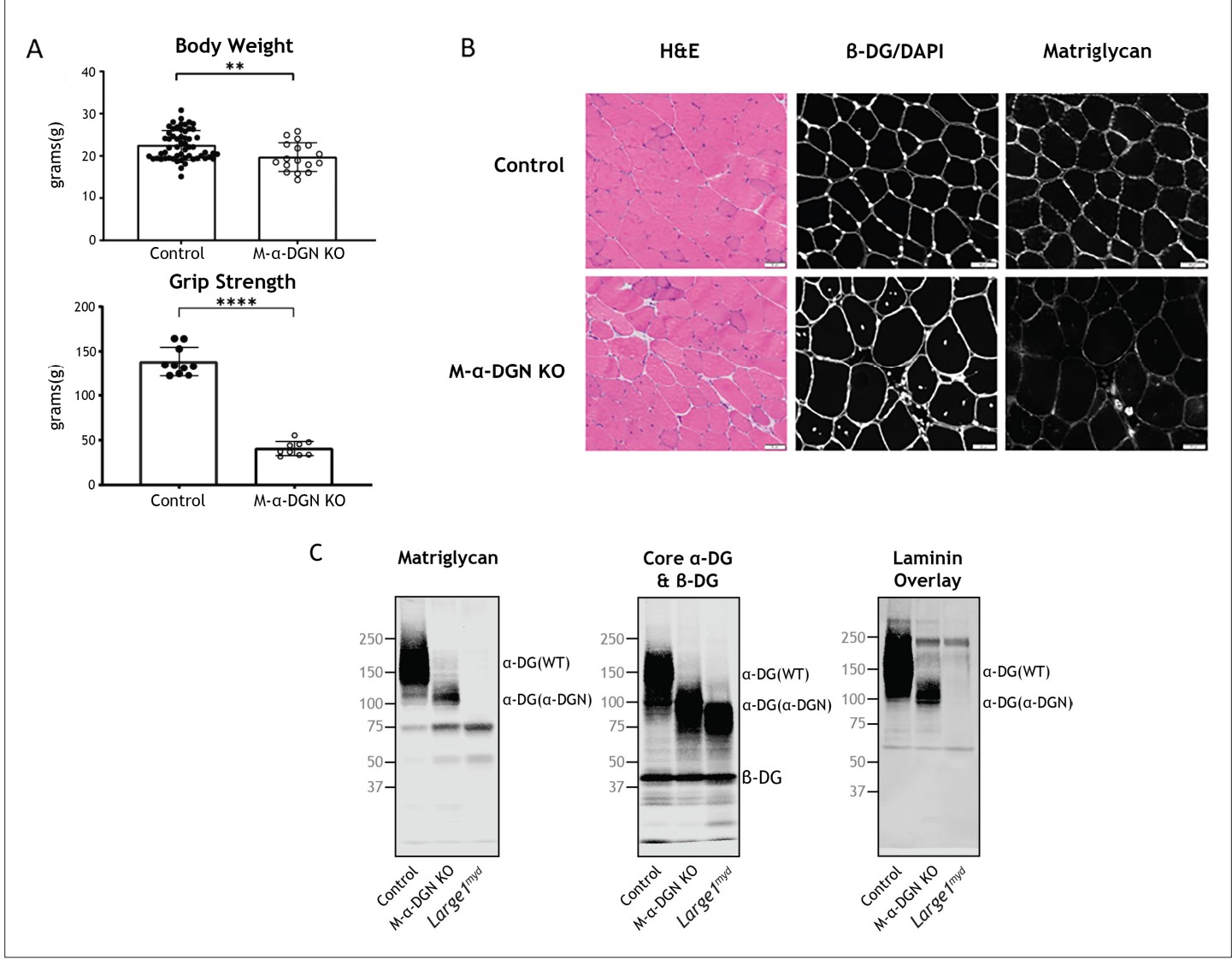

**Figure 2.** Characterization of mice with a muscle-specific loss of α-DG N-terminal (α-DGN). (**A**) Body weight and grip strength of 12-week-old wild-type (WT) littermate (control) and muscle-specific α-DGN knockout (M-α-DGN KO) mice. Double and quadruple asterisks: statistical significance determined by Student's unpaired t-test (**p-value = 0.005, ****p-value <0.0001). (**B**) Histological analyses of quadriceps muscles from 12-week-old control and M-α-DGN KO mice. Sections stained with H&E or used for immunofluorescence to detect β-DG (affinity purified rabbit anti-β-DG), DAPI, and matriglycan (IIH6). Scale bars = 50 µm. (**C**) Immunoblot analysis of skeletal muscle from control, M-α-DGN KO, and *Large1myd* mice. Glycoproteins were enriched using wheat-germ agglutinin (WGA)-agarose with 10 mM EDTA. Immunoblotting was performed to detect matriglycan (IIIH11), core α-DG, β-DG (AF6868), and laminin overlay. α-DG in WT control muscle (α-DG (WT)) and α-DG in α-DGN-deficient muscle (α-DG (Δα-DGN)) are indicated on the right. The number of KO mice was 17, and that of LC mice was 57. There were 6 male KO mice, 11 female KO mice, 23 male LC mice, and 34 female LC mice. Molecular weight standards in kilodaltons (kDa) are shown on the left.

The online version of this article includes the following source data and figure supplement(s) for figure 2:

**Source data 1.** Full blots for *Figure 2C*.

**Figure supplement 1.** Mice heterozygous (+/-) for a constitutive deletion of α-DG N-terminal (α-DGN) have two different sizes of α-DG.

**Figure supplement 1—source data 1.** Full blots for *Figure 2—figure supplement 1*.

**Figure supplement 2.** The short 100–120 kDa band in muscle-specific α-DGN knockout (M-α-DGN KO) muscle is matriglycan.

**Figure supplement 2—source data 1.** Full blots for *Figure 2—figure supplement 2*.

streaks extending beyond the pre-synaptic terminal (**Figure 3A**). Post-synaptic morphology in adult M-α-DGN KO mice was predominately irregular in the tibialis anterior (TA), extensor digitorum longus (EDL), and soleus (SOL) muscles (**Figure 3B**). Although the overall synaptic size did not differ between controls and M-α-DGN KO mice, the dispersion of AChR clusters was greater in the M-α-DGN KOs (**Figure 3B**), in line with an increased percentage of plaque-like formations and AChR extensions that projected beyond the nerve terminal. Despite the post-synaptic abnormalities, all NMJs from M-α-DGN KO mice were fully innervated.

To determine what effect the loss of α-DGN has on muscle force production, we characterized the phenotype and function of EDL muscles in 12- to 17-week-old WT (control) and M-α-DGN KO mice. Specifically, we measured muscle mass, muscle cross-sectional area (CSA), production of absolute isometric tetanic force, specific force, and lengthening contraction-induced reduction in force. Muscle mass and CSA were comparable between control and M-α-DGN KO mice (**Figure 4A and B**). Although the production of absolute isometric tetanic force was significantly lower in M-α-DGN KO mice compared to control mice (**Figure 4C**), specific forces were comparable between the two groups when normalized to muscle CSA (**Figure 4D**). Lengthening contraction-induced force reduction in M-α-DGN KO EDL remained greater than those from control EDL for the entire 60 min that muscles were assessed (**Figure 4E**). These results suggest that the short form of matriglycan on α-DG in M-α-DGN KO EDL enables force production but cannot prevent force reduction caused by lengthening contractions. POMK KO skeletal muscle also expresses a short form of matriglycan, similar to M-α-DGN KO muscle. The short form of matriglycan in POMK KO skeletal muscle maintains force production but similarly cannot prevent lengthening contraction-induced force decline (**Walimbe et al., 2020**). We, therefore, compared the function of EDL muscles from POMK KO mice with those from M-α-DGN KO mice ex vivo. We did not observe significant differences in lengthening contraction-induced force deficits between the two mouse strains (**Figure 4—figure supplement 1**). These results suggest that muscle with similar size matriglycan exhibits similar force production and lengthening contraction-induced force decline.

We next determined if exogenous DG lacking α-DGN (**Figure 5**) produces the short form of matriglycan. We first produced muscle-specific DG KO mice to achieve a muscle-specific deletion of DG by crossing mice expressing Cre under control of the *paired box 7* (*Pax7*) promoter (*Pax7^Cre^*) to *Dag1^flox/flox^* mice to generate *Pax7^Cre^; Dag1^flox/flox^* (M-*Dag1* KO) mice. To assess the presence of DG, we performed immunostaining of quadriceps muscles from 12-week-old M-*Dag1* KO mice, which showed the absence of β-DG-positive fibers (**Figure 5—figure supplement 1A**). Immunoblot analysis showed that α-DG derived from skeletal muscle was absent in M-*Dag1* KO mice (**Figure 5—figure supplement 1B**). However, consistent with prior reports, peripheral nerve-derived α-DG of 110 kDa is observed in M-*Dag1* KO mice in the presence of EDTA, which improves the extraction of matriglycan-positive α-DG (**Saito et al., 2003**). To assess muscle function, we evaluated muscle-specific force and lengthening contraction-induced reduction in force ex vivo, which showed that muscle-specific force was significantly decreased and that muscles were more susceptible to lengthening contraction-induced force decline in the absence of DG (**Figure 5—figure supplement 1C, D**). Collectively, these results show that M-*Dag1* KO mice harbor a more complete loss of DG in muscle than the previously generated mouse model that deleted DG with cre driven by the muscle creatine kinase (MCK) promotor (*MCK^Cre^; Dag1^flox/flox^*) harboring muscle-specific deletion of DG (**Cohn et al., 2002**).

We next generated an adeno-associated virus (AAV) construct of DG lacking the α-DGN (AAV-MCK DG-E; **Figure 5A**), which we injected into M-*Dag1* KO mice through the retro-orbital (RO) sinus. A previous report found that matriglycan was not produced when a similar adenoviral construct of DG lacking the α-DGN was used to infect *Dag1 KO* ES cells (**Kanagawa et al., 2004**). However, we found that matriglycan of similar size was produced in M-*Dag1* KO mice injected with AAV-MCK DG-E as in M-α-DGN KO mice (**Figure 5C**). Immunofluorescence analysis of quadriceps muscles from M-*Dag1* KO mice injected with AAV-MCK DG-E showed decreased immunoreactivity to matriglycan-positive muscle fibers but restored expression of β-DG (**Figure 5B**). We previously reported that WT DG is able to restore full-length matriglycan and its function when expressed in DG null cells or muscle (**Hara et al., 2011a**; **Hara et al., 2011b**). Immunoblot analysis of skeletal muscle from M-*Dag1* KO mice injected with AAV-MCK-DG-E showed expression of α-DG containing matriglycan around ~100–125 kDa (**Figure 5C**), which was the same size as α-DG with the short form of matriglycan in M-α-DGN KO muscle (**Figure 2C**). The molecular weight of α-DG was decreased in muscle from M-*Dag1* KO

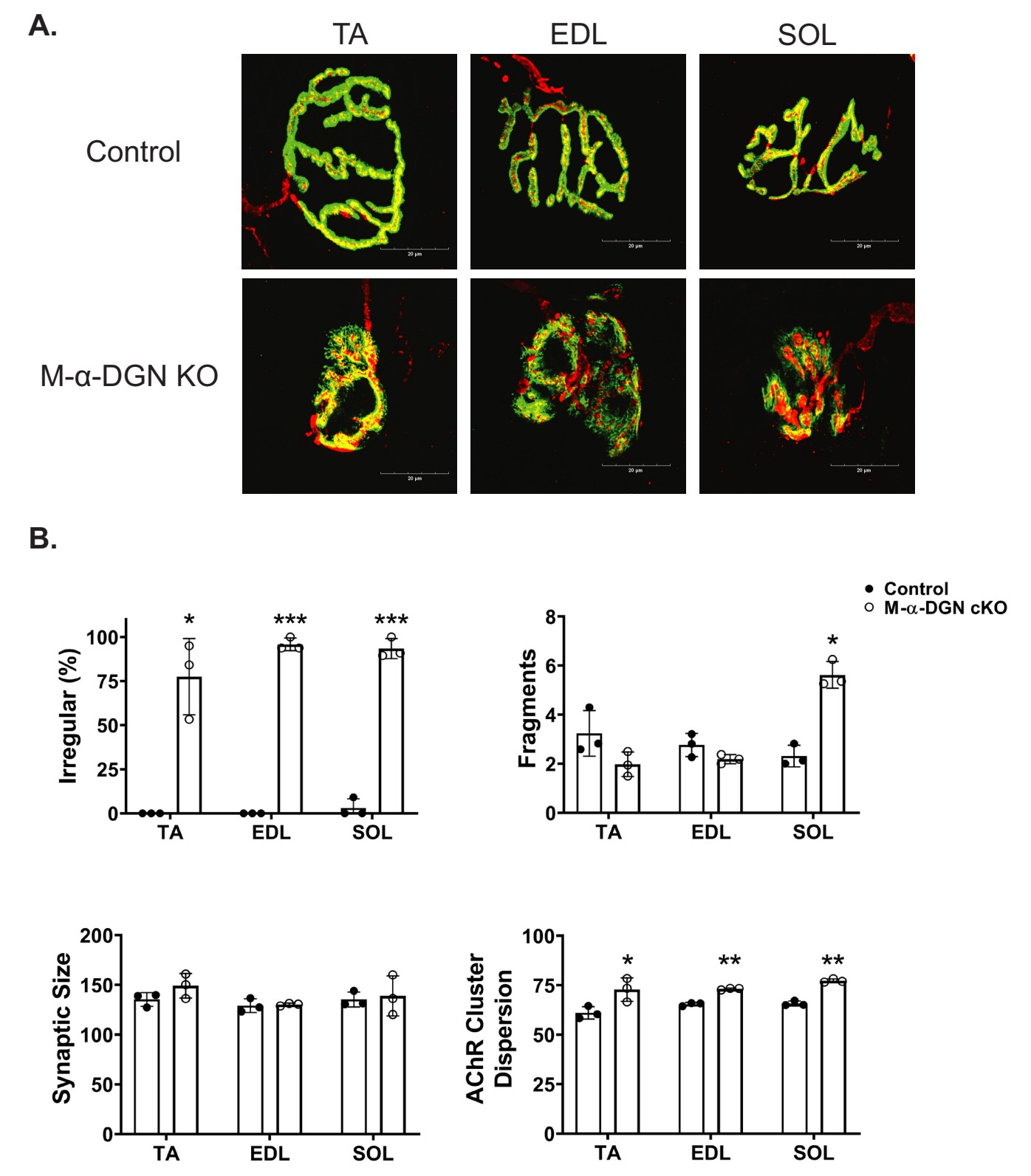

**Figure 3.** α-DG N-terminal (α-DGN) deficiency results in post-synaptic defects. Neuromuscular junctions (NMJs) from tibialis anterior (TA), extensor digitorum longus (EDL), and soleus (SOL) muscles obtained from 35- to 39-week-old control and muscle-specific α-DGN knockout (M-α-DGN KO) mice. (**A**) Representative images of post-synaptic terminals (α-BTX-488; green), motor axons (anti-neurofilament-H; red), and pre-synaptic terminals (anti-synaptophysin; red) from TA, EDL, and SOL muscles. Scale bars = 20 μm. (**B**) Scoring of post-synaptic defects by blinded observers (scoring criteria

*Figure 3 continued on next page*

*Figure 3 continued*

described in Materials and methods). Statistical significance determined by Student's unpaired t-test; *p-value <0.05; **p-value <0.001; ***p-value <0.0001.

mice, similar to that observed in M-α-DGN KO mice, whereas the molecular weight of β-DG was unchanged relative to M-α-DGN KO mice (*Figure 2C*). We also observed laminin binding at ~100–125 kDa in muscle from M-*Dag1* KO + AAV MCK DG-E mice (*Figure 5C*). In addition, we assessed the physiological effects of expressing DG without the DGN. We observed that the specific force was comparable in M-*Dag1* KO + AAV MCK DG-E and M-*Dag*1 KO mice (*Figure 5D*) and that the

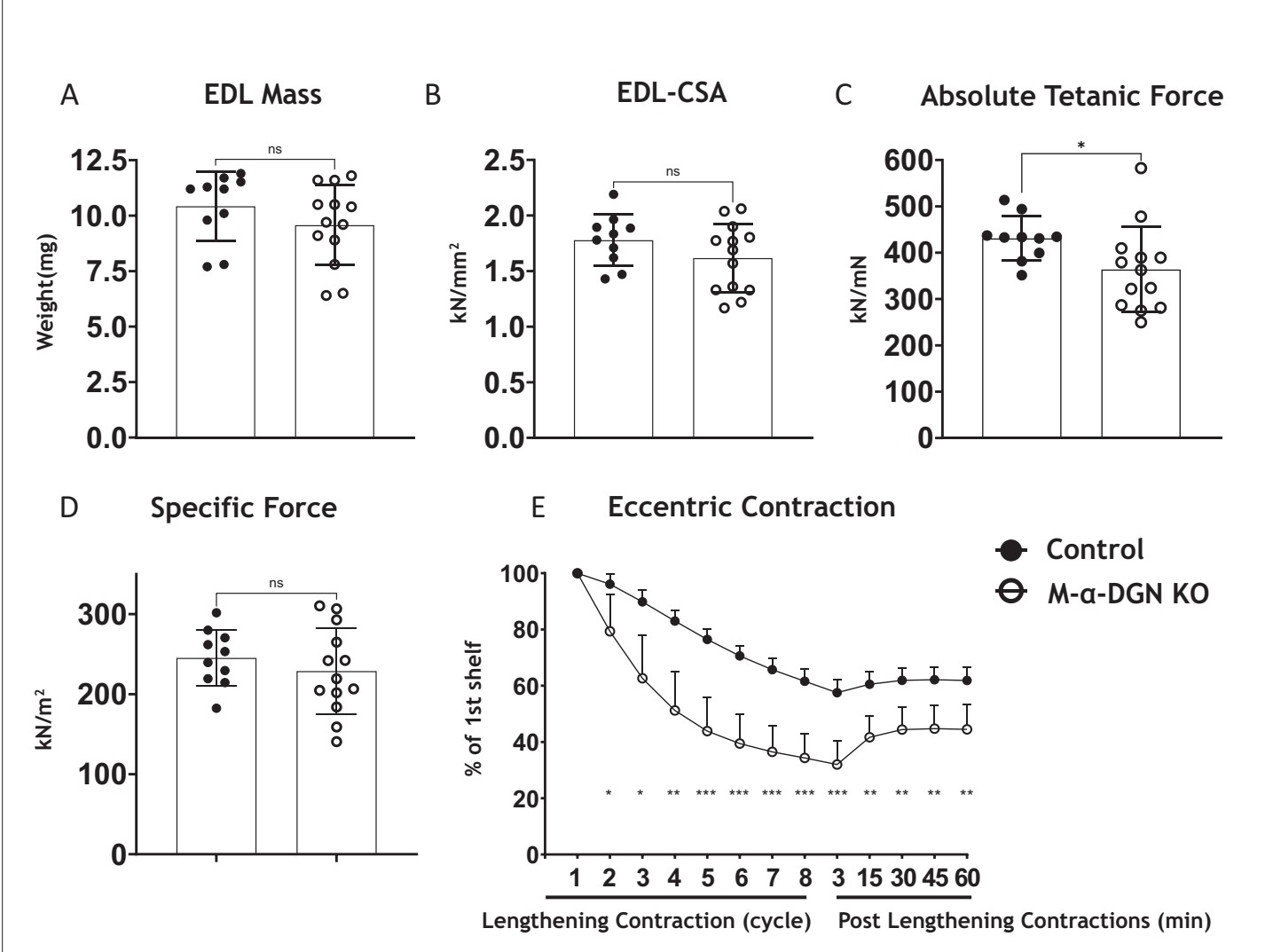

**Figure 4.** α-DG N-terminal (α-DGN)-deficient extensor digitorum longus (EDL) muscle demonstrates greater lengthening contraction-induced force decline. (**A**) Weight (mg) of EDL muscles from wild-type (WT) littermates (controls) and muscle-specific α-DGN knockout (M-α-DGN KO) mice; p=0.2469, as determined by Student's unpaired t-test. (**B**) Cross-sectional area of EDL muscles; p=0.1810, as determined by Student's unpaired t-test. (**C**) Maximum absolute tetanic force production in EDL muscles. p=0.0488, as determined by Student's unpaired t-test. (**D**) Specific force production in EDL muscles; p=0.4158, as determined by Student's unpaired t-test. (**E**) Force deficit and force recovery after eccentric contractions in EDL muscles from 12- to 17-week-old male and female control (closed circles; n=7) and M-α-DGN KO (open circles; n=7) mice. *p<0.05; **p<0.01; ***p<0.001, as determined by Student's unpaired t-test of at any given lengthening contractions cycle. Bars represent the mean ± the standard deviation.

The online version of this article includes the following figure supplement(s) for figure 4:

**Figure supplement 1.** α-DG N-terminal (α-DGN)-deficient muscle and protein *O*-mannose kinase (POMK)-deficient muscle with similar short forms of matriglycan exhibit similar lengthening contraction-induced force decline.

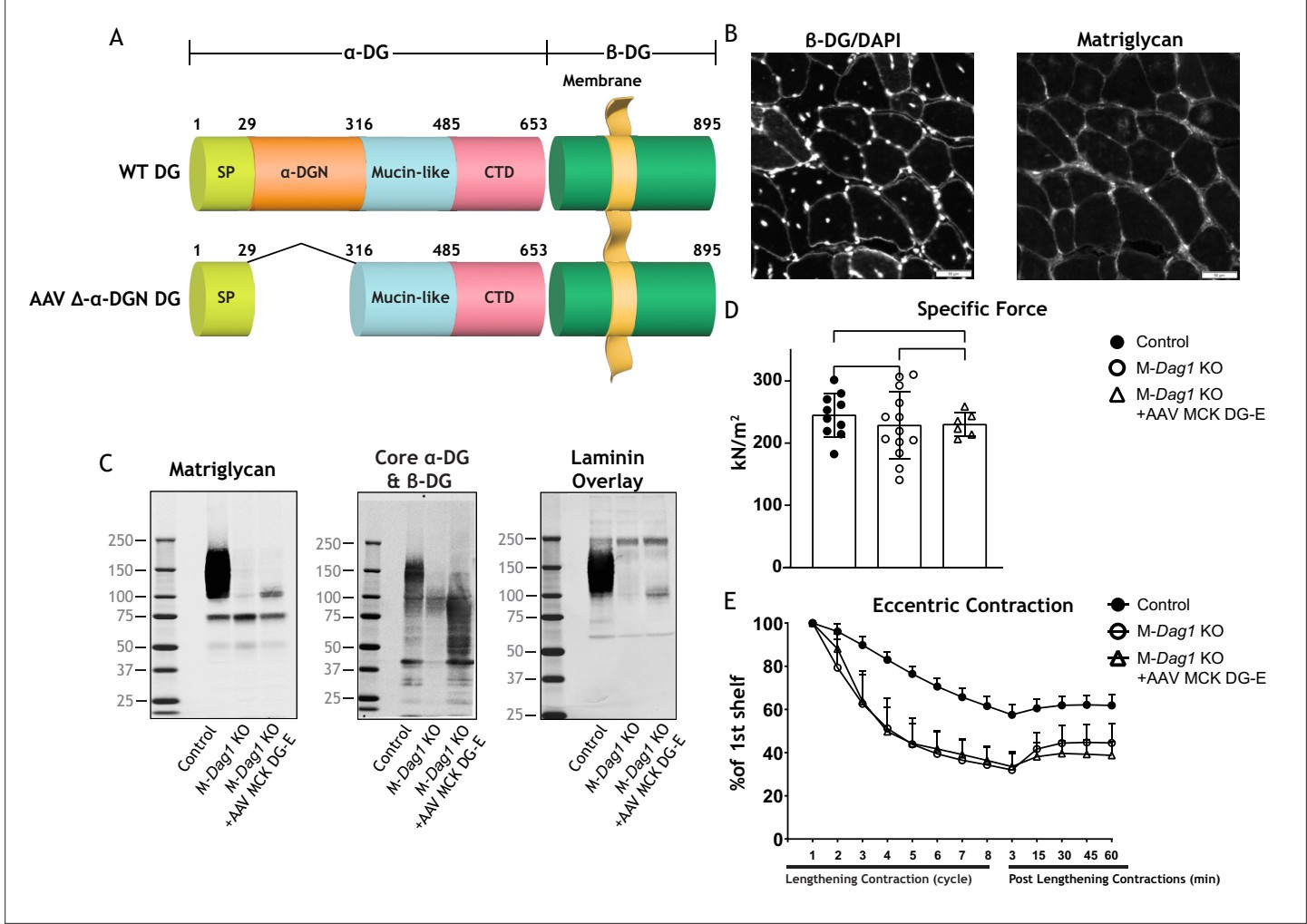

**Figure 5.** Exogenous α-DG N-terminal (α-DGN)-deficient dystroglycan (DG) also produces short matriglycan like M-*Dag1* KO muscle. (**A**) Schematic representation of a wild-type (WT) DG and an adeno-associated virus (AAV) carrying a mutant DG in which the N-terminal domain has been deleted (DG-E). α-DG is composed of a signal peptide (SP, amino acids 1–29), an N-terminal domain (amino acids 30–316), a mucin-like domain (amino acids 317–485), and a C-terminal domain (amino acids 486–653). The green box represents β-DG. (**B**) Immunofluorescence analyses of quadriceps muscles from 12-week-old M-*Dag1* KO mice injected with AAV-MCK DG-E to detect β-DG, nuclei (DAPI), and matriglycan (IIH6). Scale bars = 50 µm. (**C**) Immunoblot analysis of skeletal muscle obtained from littermate controls (control), M-*Dag1* KO mice, or M-*Dag1* KO mice injected with AAV-MCK DG-E. Glycoproteins were enriched from skeletal muscles using wheat-germ agglutinin (WGA)-agarose. Immunoblotting was performed to detect matriglycan (IIIH11), core α-DG and β-DG (AF6868), and laminin (overlay). (**D**) Production of specific force in extensor digitorum longus ( EDL) muscles from 12- to 17-week-old male and female M-*Dag1* KO mice (controls; closed circles, n=10); M-α-DGN KO mice (open circles, n=13); and M-*Dag1* KO + AAV MCK DG-E mice (open triangles, n=6). p-Values determined by Student's unpaired t-test; controls vs. M-*Dag1* KO: p=0.4158; controls vs. M-*Dag1* KO + AAV MCK DG-E: p=0.3632; M-*Dag1* KO vs. M-*Dag1* KO + AAV MCK DG-E: p=0.948. (**E**) Force deficits and recovery in EDL muscles from mice in D. There is no significant difference in M-*Dag1* KO vs. M-*Dag1* KO + AAV MCK DG-E as determined by Student's unpaired t-test at any given lengthening contraction cycle or post-lengthening contraction.

The online version of this article includes the following source data and figure supplement(s) for figure 5:

**Source data 1.** Full blots for *Figure 5C*.

**Figure supplement 1.** Characteristics of M-*Dag1* KO (*Pax7cre; Dag1flox/flox*) mice.

**Figure supplement 1—source data 1.** Full blots for *Figure 5—figure supplement 1*.

**Figure supplement 2.** Excess free α-DG N-terminal (α-DGN) interferes with like-acetylglucosaminyltransferase-1 (LARGE1) elongation of matriglycan on α-DG.

**Figure supplement 2—source data 1.** Full blots for *Figure 5—figure supplement 2*.

two groups exhibited similar amounts of lengthening contraction-induced force decline (*Figure 5E*). Therefore, these data demonstrate that AAV-mediated delivery of exogenous DG lacking the α-DGN into M-*Dag1* KO mice also produces a short-form matriglycan and such mice exhibit similar muscle function as M-α-DGN KO mice.

Our studies show that DG lacking the α-DGN expresses a short form of matriglycan; this suggests that α-DGN is necessary to produce full-length matriglycan. To test this hypothesis, we determined if matriglycan expression could be restored in mice lacking α-DGN. We injected M-α-DGN KO mice with an AAV expressing α-DGN (AAV-CMV α-DGN) and harvested the skeletal muscles of these mice 8–10 weeks after injection. H&E staining in M-α-DGN KO mice injected with AAV-CMV α-DGN was unchanged from M-α-DGN KO mice (*Figure 2B* and *Figure 6A*). Quadriceps muscles from M-α-DGN KO mice injected with AAV-CMV α-DGN showed a reduced intensity of matriglycan relative to littermate controls (*Figure 6*). Immunoblot analysis of these mice showed that α-DG had a molecular weight of ~100–125 kDa, whereas β-DG remained unchanged (*Figure 6B*). Laminin binding was observed at ~100–125 kDa in M-α-DGN KO skeletal muscle infected with AAV-CMV α-DGN (*Figure 6B*). Collectively, this phenotype is similar to that observed in the skeletal muscles of M-α-DGN KO mice. Expressing α-DGN in M-α-DGN KO mice did not alter specific force or improve force deficits induced by lengthening contractions (*Figure 6C and D*). Thus, supplementing M-α-DGN KO skeletal muscle with α-DGN fails to produce full-length matriglycan. To test if excessive free α-DGN interferes with the binding of endogenous α-DGN to LARGE1, we analyzed immunoblots of skeletal muscle from control mice and those overexpressing free α-DGN (α-DGN transgenic [α-DGN Tg]). Immunoblot analysis of these mice showed that α-DG had a reduced molecular weight of ~100–125 kDa, whereas β-DG remained unchanged (*Figure 5—figure supplement 1*). This biochemical phenotype is similar to that observed in the skeletal muscles of M-α-DGN KO mice and demonstrates that free α-DGN interferes with the elongation of matriglycan, but that LARGE1 can still synthesize a short form of matriglycan on α-DG.

To determine if excess LARGE1 produces extended matriglycan in M-α-DGN KO muscle, we analyzed immunoblots of skeletal muscle from littermate controls, M-α-DGN KO and M-α-DGN KO mice injected with AAV-MCK-*Large1*. M-α-DGN KO mice injected with AAV-MCK-*Large1* demonstrated no change in the molecular weight of α-DG and β-DG relative to M-α-DGN KO. A laminin overlay using laminin-111 also showed no change (*Figure 6—figure supplement 1*). These results indicate that even if LARGE1 is overexpressed, full-length matriglycan cannot be produced without α-DGN.

The ability of matriglycan to bind ECM ligands correlates with its length (*Goddeeris et al., 2013*). We hypothesized that the susceptibility to force decline after lengthening contractions would differ depending on the length of matriglycan. To test this hypothesis, we performed physiological muscle tests in three mouse models with different size α-DG to determine the difference in susceptibility to lengthening contraction-induced force reduction. Specifically, we used: (1) M-α-DGN KO mice, which express a short form of matriglycan, (2) *Dag1^T190M^* mice, which harbor a knock-in mutation (T190M) in *Dag1* that inhibits the DG-LARGE1 interaction and leads to incomplete post-translational modification of α-DG (*Hara et al., 2011a*), and (3) C57BL/6J WT (C57) mice, which express full-length matriglycan. The percent deficit value of the eight EC shows the largest difference in the EC protocol; we, therefore, compared these values between our three different mouse models (*Figure 7A*). M-α-DGN KO mice showed a significantly higher percent deficit (70.2%±5.7) compared to C57 (41.7%±8.0) and *Dag1^T190M^* (41.6%±6.7) mice, with no difference observed between the latter groups. Laminin overlay assay in skeletal muscle showed α-DG laminin binding at ~150–250 kDa in skeletal muscle from C57 mice, ~100–150 kDa in skeletal muscle from *Dag1^T190M^* mice, and ~100–125 kDa in skeletal muscle from M-α-DGN KO mice (*Figure 7C*). Moreover, the percentage of centrally nucleated fibers differed significantly in *Dag1^T190M^* (1.73%±0.31) and M-α-DGN KO (9.28%±2.41) mice compared to C57 mice (1.22%±0.15) (*Figure 7B*). The reduction of laminin-binding activity of α-DG is thought to be the main cause of dystroglycanopathy (*Kanagawa et al., 2009*; *Goddeeris et al., 2013*). Indeed, we observed a reduced binding capacity (relative $B_{max}$) for laminin-111 in solid-phase binding analyses in skeletal muscle from M-α-DGN KO and *Dag1^T190M^* mice compared to skeletal muscle from C57 mice (10.7-fold and 2.3-fold difference relative to WT, respectively) (*Figure 7D*). However, the binding capacity of skeletal muscle from *Dag1^T190M^* and M-α-DGN KO mice was higher than that of *Large1^myd^* muscle (*Figure 7E*). M-α-DGN KO and *Dag1^T190M^* also displayed a greater dissociation constant (*Figure 7E*).

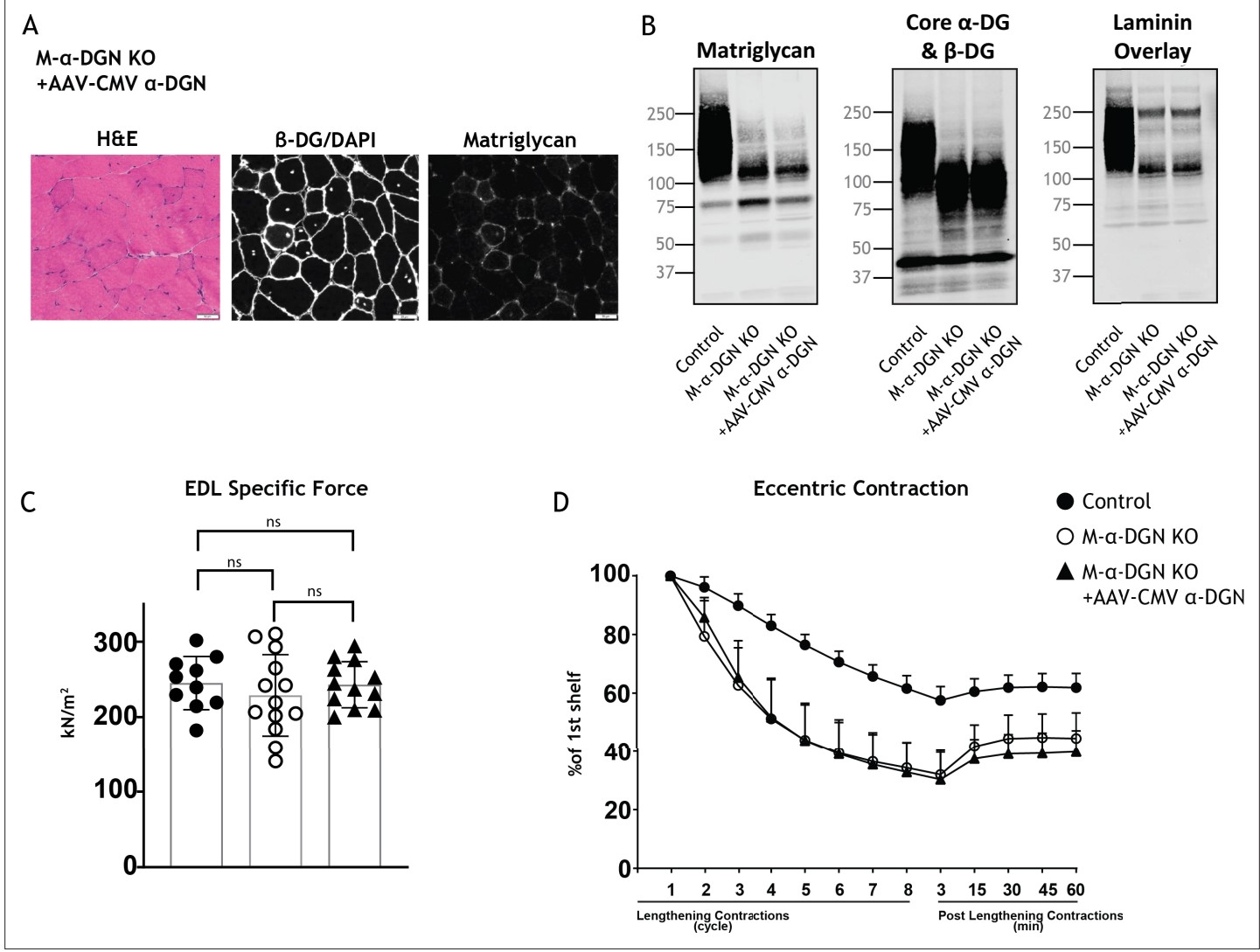

**Figure 6.** Expression of α-DG N-terminal (α-DGN) in muscle-specific α-DGN knockout (M-α-DGN KO) mice does not rescue matriglycan elongation. (**A**) Representative sections of quadriceps muscles from 17-week-old M-α-DGN KO mice injected with AAV-CMV α-DGN. Sections were stained with H&E and immunofluorescence to detect matriglycan (IIH6) and β-DG (AP83). Scale bars = 50 μm.(**B**) Immunoblot analysis of skeletal muscle obtained from littermate controls or M-α-DGN KO mice and M-α-DGN KO mice injected with AAV-CMV α-DGN (M-α-DGN KO+AAV CMV α-DGN). Glycoproteins were enriched using wheat-germ agglutinin (WGA)-agarose with 10 mM EDTA. Immunoblotting was performed to detect matriglycan (IIIH11), core α-DG and β-DG (AF6868), and laminin overlay. (**C**) Production of specific force in extensor digitorum longus (EDL) muscles from 12- to 17-week-old male and female M-α-DGN wild-type (WT) littermates (controls; closed circles, n=10); M-α-DGN KO (open circles, n=13); and M-α-DGN KO+AAV CMV α-DGN (closed triangles, n=12). p-Values determined by Student's unpaired t-test; controls vs. M-α-DGN KO+AAV CMV α-DGN: p=0.8759; controls vs. M-α-DGN KO: p=0.4333; M-α-DGN KO vs. M-α-DGN KO+AAV CMV α-DGN: p=0.4333. (**D**) Force deficit and force recovery after lengthening contractions in EDL muscles from 12- to 17-week-old male and female M-α-DGN KO WT littermates (controls, closed circles; n=6) and M-α-DGN KO (KO, open circles; n=7) mice, and in M-α-DGN KO mice injected with AAV-CMV α-DGN (KO+AAV CMV α-DGN, closed triangles; n=8). There is no significant difference in M-α-DGN KO vs. M-α-DGN KO+AAV CMV α-DGN as determined by Student's unpaired t-test at any given lengthening contractions cycle or post-lengthening contractions.

The online version of this article includes the following source data and figure supplement(s) for figure 6:

**Source data 1.** Full blots for *Figure 6B*.

**Figure supplement 1.** Like-acetylglucosaminyltransferase-1 (LARGE1) overexpression does not extend matriglycan on dystroglycan (DG) lacking α-DG N-terminal (α-DGN).

**Figure supplement 1—source data 1.** Full blots for *Figure 6—figure supplement 1*.

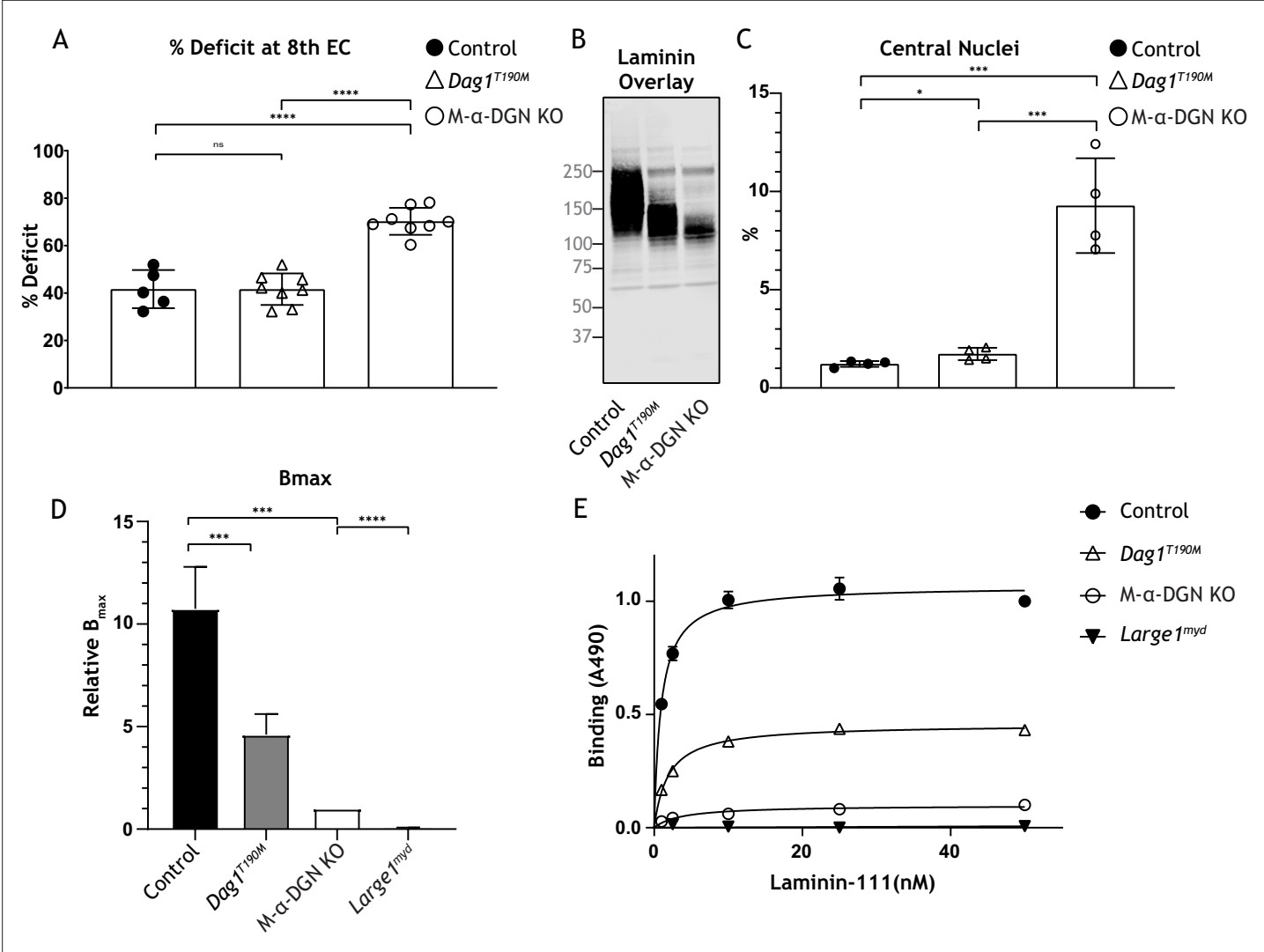

**Figure 7.** Relationship between matriglycan length and dystrophic phenotype. (**A**) Percentage deficit of eight eccentric contraction (EC) in extensor digitorum longus (EDL) muscles from C57BL/6J wild-type (WT) (control), muscle-specific α-DGN knockout (M-α-DGN KO), and *Dag1^{T190M}* mice. p-Values determined by Student's unpaired t-test; control vs. *Dag1^{T190M}*: p=0.0263; control and *Dag1^{T190M}* vs. M-α-DGN KO: p<0.001. (**B**) Immunoblot analysis of quadriceps skeletal muscles from control, *Dag1^{T190M}* and M-α-DGN KO mice. Glycoproteins were enriched using wheat-germ agglutinin (WGA)-agarose with 10 mM EDTA. Immunoblotting was performed with laminin (laminin overlay). (**C**) Percentage of muscle fibers with central nuclei in 12- to 19-week-old control, *Dag1^{T190M}* and M-α-DGN KO mice; n=4 for all groups. p-Values determined by Student's unpaired t-test; control and *Dag1^{T190M}* vs. M-α-DGN KO: p<0.001; control vs. *Dag1^{T190M}*: p=0.0263. (**D**) Comparison of average solid-phase determined relative $B_{max}$ values for laminin. $B_{max}$ values for M-α-DGN KO were set to 1 to allow for direct comparisons; error bars indicate s.e.m. p-Values determined using Student's unpaired t-test; control vs. *Dag1^{T190M}* and control vs. M-α-DGN KO: p<0.01, and M-α-DGN KO vs. *Large1^{myd}*: p<0.001. (**E**) Solid-phase analysis of laminin-binding using laminin-111 in skeletal muscle from control, *Dag1^{T190M}*, M-α-DGN KO, and *Large1^{myd}* KO mice (three replicates for each group). Control $K_d$: 0.9664±0.06897 nM; *Dag1^{T190M}* $K_d$: 1.902±0.1994 nM; and M-α-DGN KO $K_d$: 2.322±0.6114 nM.

The online version of this article includes the following source data for figure 7:

**Source data 1.** Full blot for *Figure 7B*.

Collectively, these results suggest that matriglycan-positive α-DG of at least ~150 kDa is sufficient to prevent force decline from lengthening contractions and significant dystrophic changes, despite a 45% reduction in laminin-binding activity.

## Discussion

Matriglycan is a scaffold for LG domain-containing ECM proteins (e.g., laminin, agrin, and perlecan). During skeletal muscle differentiation, matriglycan is elongated by LARGE1, increasing its binding capacity for ECM ligands, which allows it to serve as a matrix scaffold that is required for skeletal muscle function. Consequently, defects in matriglycan synthesis led to dystroglycanopathies, such as congenital and limb-girdle muscular dystrophies that can be accompanied by structural brain and eye abnormalities. Notably, there is a direct correlation between the number of GlcA-Xyl repeats on α-DG and its binding capacity for ECM ligands. Indeed, patients with increased clinical severity of dystroglycanopathy have few GlcA-Xyl repeats and a shorter form of matriglycan. Despite the importance of mature, full-length matriglycan for proper muscle structure and function, little is known about mechanisms that control its elongation.

We now show that LARGE1 synthesizes a non-elongated form of matriglycan on α-DG lacking α-DGN (i.e., α-DGN-deleted DG). This short form of matriglycan binds laminin and maintains specific force. However, it fails to prevent reduced force production after ECs, NMJ abnormalities, or dystrophic changes in muscle. Collectively, our study shows that LARGE1 requires α-DGN to generate full-length matriglycan in skeletal muscle and thus prevents muscle pathophysiology. However, synthesis of a shorter form of matriglycan can proceed independently of the α-DGN domain.

Our results suggest that the interaction between LARGE1 and α-DGN holds the enzyme-substrate complex together over multiple cycles of sugar addition. These findings build on our previous study demonstrating that phosphorylation of the core M3 trisaccharide by POMK is also necessary for matriglycan elongation (*Walimbe et al., 2020*). Thus, the generation of full-length matriglycan on α-DG (~150–250 kDa) by LARGE1 requires both POMK and α-DGN; in the absence of either, LARGE1 synthesizes a shorter form of matriglycan. Thus, the interaction between LARGE1 and α-DG is essential for efficient matriglycan polymerization. Although deep-learning tools could be used to predict protein-protein interactions, it is unlikely that these tools will be able to predict the interaction between LARGE1 and DG as this requires post-translational modifications (*Walimbe et al., 2020*).

In our study, muscle-specific deletion of α-DGN resulted in the production of short forms of matriglycan on α-DG (~100–125 kDa). Mice lacking α-DGN exhibited low body weight and grip strength, and histological characterization of quadriceps muscles revealed mild muscular dystrophy and a lack of homogeneous matriglycan expression. Physiological examination revealed that M-α-DGN KO muscle was susceptible to lengthening contraction-induced force decline, although specific force was maintained. These results are consistent with those obtained when α-DGN-deleted DG was expressed in muscle-specific DG null mice and confirm that α-DG lacking α-DGN produces short forms of matriglycan, which do not prevent dystrophic muscle changes in this mouse model. Furthermore, DG in the post-synaptic membrane is known to play a key role in synaptic maturation (*Nishimune et al., 2008*). However, the NMJs in M-α-DGN KO mice in our study were abnormal and irregularly shaped. This indicates both that DG and full-length matriglycan are required for synaptic maturation.

If LARGE1 binding to α-DGN enables its ability to elongate matriglycan, then we would expect that rescuing M-α-DGN KO skeletal muscle with α-DGN would restore the expression of full-length matriglycan. However, this failed to occur and suggests that restoring α-DGN expression alone is not sufficient for LARGE1 to elongate matriglycan. These results indicate that the ability of LARGE1 to elongate matriglycan requires α-DG with α-DGN. This finding is consistent with data showing that matriglycan is not elongated when α-DGN is deleted, even when LARGE1 is overexpressed. Therefore, α-DGN acts as a recognition site for the glycosyltransferase LARGE1 and establishes a model where α-DGN, together with phosphorylated core M3, anchors LARGE1 to the matriglycan production site to enable its synthesis and elongation. Notably, although the molecular recognition of α-DGN by LARGE1 is considered essential for the expression of functional α-DG (*Kanagawa et al., 2004*), our results show that LARGE1 can synthesize a short non-elongated form of matriglycan in the absence of α-DGN, indicating that LARGE1 does not need to interact with α-DGN to function.

To determine how much matriglycan is needed to prevent lengthening contraction-induced reductions in force, we studied mice that express different sizes of matriglycan. Muscle from M-α-DGN KO

mice showed an increased force deficit and a 7.6-fold increase in centrally nucleated fibers compared to muscle from C57 mice, indicating that the short form of matriglycan does not prevent dystrophic changes. However, despite the lower amount of matriglycan in muscle from $Dag1^{T190M}$ mice compared to that from C57 mice, the force deficit was not different between the two groups, and centrally nucleated fibers were increased in α-DGN mutant ($Dag1^{T190M}$) mice by only 1.4-fold. This indicates that a matriglycan-positive α-DG over 150 kDa can prevent muscular dystrophy.

Muscular dystrophy is not observed in a mouse model of Fukuyama congenital muscular dystrophy, which occurs due to a retrotransposition insertion in the mouse *fukutin* ortholog and leads to laminin binding at 50% of normal levels (*Kanagawa et al., 2009*). In $Dag1^{T190M}$ mice, the laminin-binding level is about 45% of normal, which likely explains the mild increase in centrally nucleated fibers compared to muscle from C57 mice. However, in muscle from M-α-DGN KO mice, the laminin-binding level is only 9% relative to that of C57 mice and leads to a marked increase in centrally nucleated fibers and force deficit induced by lengthening contractions. This indicates that matriglycan length is critical for regulating damage induced by lengthening contractions and that the production of ~120–150 kDa α-DG significantly prevents dystrophic change, suggesting that this pathologic effect can be prevented without the expression of full-length matriglycan. Thus, our results describe a relationship between matriglycan size, damage induced by lengthening contractions, and the degree of dystrophic change.

α-DGN is cleaved by the proprotein convertase furin at the sequence RVRR (amino acids 309–312) (*Singh et al., 2004*). It has a globular structure that is organized into two subdomains (*Brancaccio et al., 1997*). The first subdomain is a typical Ig-like domain; the second subdomain resembles ribosomal RNA-binding proteins (*Bozic et al., 2004*). α-DGN is secreted by cells in culture (*Saito et al., 2011*) and has been detected in a wide variety of human bodily fluids, including serum and plasma (*Saito et al., 2008*; *Saito et al., 2011*). Decreased levels of α-DGN have been reported in the serum of patients with Duchenne muscular dystrophy (DMD) and in the serum of utrophin-deficient mdx mice, a mouse model for DMD (*Crowe et al., 2016*). Finally, we have also studied the protective role of secreted α-DGN (*de Greef et al., 2019*) on influenza A virus proliferation.

Collectively our study demonstrates that synthesis of full-length matriglycan on α-DG (~150–250 kDa) requires α-DG containing α-DGN. In the absence of α-DGN, LARGE1 can synthesize a short non-elongated form of matriglycan on α-DG (~100–125 kDa) in skeletal muscle in a process that does not require that LARGE1 interacts with α-DGN. Taken together, these findings demonstrate that matriglycan length regulates the severity of muscular dystrophy. Interestingly, α-DG of approximately 100–125 kDa was seen in muscle from mild FKRP patients (*Brockington et al., 2001*). Our work, thereby, enhances our understanding of the mechanisms underlying matriglycan synthesis and further identifies this pathway as a possible therapeutic target for the treatment of α-dystroglycanopathy.

## Materials and methods

### Animals

All mice were maintained in a barrier-free, specific pathogen-free grade facility and had access to normal chow and water ad libitum. All animals were manipulated in biosafety cabinets and change stations using aseptic procedures. The mice were maintained in a climate-controlled environment at 25°C on a 12/12 hr light/dark cycle. Animal care, ethical usage, and procedures were approved and performed in accordance with the standards set forth by the National Institutes of Health and the University of Iowa Animal Care and Use Committee (IACUC). Mouse lines used in the study that have been previously described are: $Dag1^{-/-}$ (JAX# 006836; *Williamson et al., 1997*), $Dag1^{flox}$ (JAX# 009652; *Cohn et al., 2002*), $Dag1^{\Delta\alpha\text{-}DGN}$ (*de Greef et al., 2019*), $Dag1^{T190M}$ (*Hara et al., 2011a*), $Large1^{myd}$ (JAX# 000300) (*Lane et al., 1976*), $Mck^{cre}$ (JAX# 006475) (*Brüning et al., 1998*), $Pax7^{cre}$ (JAX# 010530) (*Keller et al., 2004*), and $Mck^{cre} Pax7^{cre} POMK^{flox}$ (*Walimbe et al., 2020*). Littermate controls were employed whenever possible. The number of animals required and the number of replicants performed were based on previous studies (*de Greef et al., 2016*; *Goddeeris et al., 2013*; *Walimbe et al., 2020*) and experience with standard deviations of the given techniques. The number of mice used is provided in each figure legend. No outliers were encountered, and no mice were excluded.

## Mouse lines

### Muscle-specific DG knockout mice (*Pax7*cre; *Dag1*flox/flox)

Male mice expressing the *Pax7*$^{Cre}$ transgene were bred to female mice that were homozygous for the floxed *Dag1* allele (*Dag1*$^{flox/flox}$). Male F1 progeny with the genotype *Pax7*$^{Cre}$; *Dag1*$^{flox/+}$ were bred to female *Dag1*$^{flox/flox}$ mice. A *Cre* PCR genotyping protocol was used to genotype the *Cre* allele using standard *Cre* primers. The primers used were Sense: TGATGAGGTTCGCAAGAACC and Antisense: CCATGAGTGAACGAACCTGG. Genotyping of *Pax7*$^{Cre}$; *Dag1*$^{flox/flox}$ mice was performed by Transnetyx using real-time PCR.

### M-α-DGN KO mice

Male mice expressing the *Pax7*$^{Cre}$ transgene were bred to female mice that were heterozygous for the *Dag1*$^{\Delta\ \alpha\text{-}DGN}$ allele (*Dag1*$^{wt/\Delta\alpha\text{-}DGN}$). Male F1 progeny with the genotype *Pax7*$^{Cre}$; *Dag1*$^{wt/\Delta\alpha\text{-}DGN}$ were bred to female mice homozygous for the floxed *Dag1* allele (*Dag1*$^{flox/flox}$). Genotyping of *Pax7*$^{Cre}$; *Dag1*$^{flox/\Delta\alpha\text{-}DGN}$ mice was performed by Transnetyx using real-time PCR. For studies with M-α-DGN KO mice, three mice of each genotype (control and *Pax7*$^{Cre}$; *Dag1*$^{flox/\Delta\alpha\text{-}DGN}$) were used.

### Generation of α-DGN Tg mice

Transgenic mice, which overexpressed only the α-DGN domain of DG, were generated using C57BL/6NJ mice. First, total RNA was extracted from mouse skeletal muscle to prepare a construct, and RT-PCR obtained the full-length cDNA of the *Dag1* gene containing the Kozak sequence. Using this template, we amplified bases 1–930 of *Dag1* corresponding to α-DGN by PCR and inserted a stop codon at the 3' end. The PCR product was cut with restriction enzymes and cloned into an EcoR I/Not I site of plasmid pCAGGS (Unitech). In addition, a fragment containing the CAG promoter and *Dag1* 1–930 was cut out from the same vector using Sal 1/Hind III and inserted into a vector for microinjection. The vector was microinjected into a pronuclear stage of zygotes of WT mice and transferred to pseudopregnant females. Offspring were screened for genomic integration of this fragment by PCR of tail DNA using the following unique sequence of the vector: forward 5'-CCTACAGCTCCTGGGC AACGTGCTGGTT-3', reverse 5'-AGAGGGAAAAAGATCTCAGTGGTAT-3'. Mice were generated by breeding F1 heterozygous transgenic males to WT females.

### Forelimb grip strength test

Forelimb grip strength was measured at 3 months using previously published methods (*de Greef et al., 2016*; *Walimbe et al., 2020*). A mouse grip strength meter (Columbus Instruments, Columbus, OH, USA) was mounted horizontally, with a non-flexible grid connected to the force transducer. The mouse was allowed to grasp the grid with its two front paws and then pulled away from the grid by its tail until the grip was broken. This was done three times over five trials, with a 1 min break between each trial. The gram force was recorded per pull, and any pull where only one front limb or any hind limbs were used was discarded. If the mouse turned, the pull was also discarded. After 15 pulls (five sets of three pulls), the mean of the three highest pulls of the 15 was calculated and reported. Statistics were calculated using GraphPad Prism 8 software. Student's t-test was used (two-sided). Differences were considered significant at a p-value less than 0.05. Graph images were also created using GraphPad Prism and the data in the present study are shown as the means ± SD unless otherwise indicated.

### Body weight measurements

Mice were weighed as previously described (*de Greef et al., 2016*; *Walimbe et al., 2020*). Weights were measured after testing grip strength using a Scout SPX222 scale (OHAUS Corporation, Parsippany, NJ, USA), and the tester was blinded to genotype. Statistics were calculated using GraphPad Prism 8 software and Student's t-test was used (two-sided). Differences were considered significant at a p-value less than 0.05. Graph images were also created using GraphPad Prism and the data in the present study are shown as the means ± SD unless otherwise indicated.

## Measurement of in vitro muscle function

To compare the contractile properties of muscles, EDL muscles were surgically removed as described previously (*Rader et al., 2016*; *de Greef et al., 2016*; *Walimbe et al., 2020*). The muscle was immediately placed in a bath containing a buffered physiological salt solution (composition in mM: NaCl, 137; KCl, 5; CaCl$_2$, 2; MgSO$_4$, 1; NaH$_2$PO$_4$, 1; NaHCO$_3$, 24; glucose, 11). The bath was maintained at 25°C, and the solution was bubbled with 95% O$_2$ and 5% CO$_2$ to stabilize pH at 7.4. The proximal tendon was clamped to a post and the distal tendon was tied to a dual mode servomotor (Model 305C; Aurora Scientific, Aurora, ON, Canada). Optimal current and whole muscle length (L$_0$) were determined by monitoring isometric twitch force. Optimal frequency and maximal isometric tetanic force (F$_0$) were also determined. The muscle was then subjected to an EC protocol consisting of eight ECs at 3 min intervals. A fiber length L$_f$-to-L$_0$ ratio of 0.45 was used to calculate L$_f$. Each EC consisted of an initial 100 ms isometric contraction at optimal frequency immediately followed by a stretch of L$_o$ to 30% of L$_f$ beyond L$_o$ at a velocity of 1 L$_f$/s at optimal frequency. The muscle was then passively returned to L$_o$ at the same velocity. At 3, 15, 30, 45, and 60 min after the EC protocol, isometric tetanic force was measured. After the analysis of the contractile properties, the muscle was weighed. The CSA of muscle was determined by dividing the muscle mass by the product of L$_f$ and the density of mammalian skeletal muscle (1.06 g/cm$^3$). The specific force was determined by dividing F$_o$ by the CSA (kN/mm$^2$). Eighteen- to 20-week-old male mice were used, and right and left EDL muscles from each mouse were employed whenever possible, with five to eight muscles used for each analysis. Each data point represents an individual EDL. Statistics were calculated using GraphPad Prism 8 software and Student's unpaired t-test was used (two-sided). Differences were considered significant at a p-value less than 0.05.

## H&E and immunofluorescence analysis of skeletal muscle

Histology and immunofluorescence of mouse skeletal muscle were performed as described previously (*Goddeeris et al., 2013*). Mice were euthanized by cervical dislocation and directly after sacrifice, quadriceps muscles were isolated, embedded in OCT compound, and then snap-frozen in liquid nitrogen-cooled 2-methylbutane. Ten μM sections were cut with a cryostat (Leica CM3050S Research Cryostat; Amsterdam, The Netherlands) and H&E stained using conventional methods. Whole digital images of H&E-stained sections were taken by a VS120-S5-FL Olympus slide scanner microscope (Olympus Corporation, Tokyo, Japan). For immunofluorescence analyses, a mouse monoclonal antibody to matriglycan on α-DG (IIH6, 1:100 dilution, Developmental Studies Hybridoma Bank, University of Iowa; RRID:AB_2617216) was added to sections overnight at 4°C followed by Alexa Fluor-conjugated goat IgG against mouse IgM (Invitrogen, Carlsbad, CA, USA, 1:500 dilution) for 40 min. The sections were also stained with rabbit polyclonal antibody to β-DG (AP83; 1:50 dilution) followed by Alexa Fluor-conjugated 488 Goat anti-rabbit IgG (1:500). Whole sections were imaged with a VS120-S5-FL Olympus slide scanner microscope. Antibody IIH6 is a mouse monoclonal to matriglycan on α-DG (*Ervasti and Campbell, 1993*), and AP83 is a rabbit polyclonal antibody to the C-terminus of β-DG (*Ervasti and Campbell, 1993*), both of which have been described previously.

For histologic analysis of skeletal muscle, H&E staining on 10 μM frozen section was performed using the Leica ST5020 Multistainer workstation (Leica Biosystems, Buffalo Grove, IL, USA) according to the manufacturer's instructions. For immunofluorescence analysis, unfixed frozen serial sections (7 μM) were incubated with primary antibodies for 1 hr, and then with the appropriate biotinylated secondary antibodies for 30 min followed by streptavidin conjugated to Alexa Fluor 594 (Thermo Fisher Scientific, UK) for 15 min. Primary antibodies used were mouse monoclonal: α-DG IIH6 (clone IIH6C4) (*Ervasti and Campbell, 1993*), β-DG (Leica, Milton Keynes, UK; clone 43DAG1/8D5). All washes were made in PBS and incubations were performed at room temperature. Sections were evaluated with a Leica DMR microscope interfaced to MetaMorph (Molecular Devices, Sunnyvale, CA, USA).

## NMJ morphology

Immediately upon harvest, EDL muscles were washed in PBS three times for 5 min each. EDL muscles were fixed in 4% paraformaldehyde for 20 min followed by three washes in PBS. Fixed muscle samples were split into three to four fiber bundles before incubating in 3% Triton X-100/PBS for 3 hr at 4°C. Muscles were subsequently washed in PBS followed by blocking at 4°C for

4 hr in Background Buster (Innovex; NB306). Samples incubated with primary antibodies against neurofilament H (NF-H; EnCor; CPCA-NF-H) at 1:1000 and synaptophysin (Thermo Fisher Scientific; MA5-14532) at 1:100 diluted in 5% Background Buster/1% Triton X-100/PBS at 4°C overnight. The muscles were then washed with PBS and incubated with fluorescently conjugated secondary antibodies and Alexa Fluor 488-conjugated α-bungarotoxin (Invitrogen; B13422) diluted in 5% Background Buster/PBS for 2 hr. Images were acquired using an Olympus FLUOVIEW FV3000 confocal laser scanning microscope. Complete enface NMJs were identified and acquired with Z-stacks using 60× and 100× objectives. Maximum intensity Z-stacks were reconstructed with the FV31S (Olympus) software and deconvoluted with cellSens Dimension (Olympus). Blinded observers analyzed α-BTX-488-labeled AChR cluster formations to determine irregularities, fragmentation, synaptic size, and dispersion. Irregularities included AChR plaques, AChR perforated plaques, ring-shaped or c-shaped clusters, and extensive fragmentation. Fragmentation was determined by the number of identifiable individual AChR clusters within the footprint of the synapse. Fiji ImageJ software was used for semi-automatic analysis of AChR clusters. Synaptic size refers to the total perimeter or footprint of the postsynapse. AChR cluster dispersion was determined by the (total stained area/total area) *100.

## Tissue biochemical analysis

Mouse skeletal muscle was minced into small pieces and homogenized with polytron (Kinematica, PT10-35) three times for 10 s at power 4–5 in 15 mL of TBS (150 mM NaCl) with 1% TX-100 and 10 mM EDTA, and protease inhibitors (per 10 mL buffer: 67 mL each of 0.2 M phenylmethylsulfonylfluoride (PMSF), 0.1 M benzamidine, and 5 µL of each of leupeptin (Sigma/Millipore) 5 mg/mL, pepstatin A (Millipore) 1 mg/mL in methanol, and aprotinin (Sigma-Aldrich) 5 mg/mL. The samples were incubated in a cold room for 1 hr with rotation. The samples were centrifuged in a Beckman Coulter Avanti J-E centrifuge for 30 min at 20,000× $g$, 4°C. The supernatant was combined with WGA slurry at 600 µL per gram of starting muscle and rotated at 4°C overnight.

The WGA beads were washed using 10× volume of WGA beads/wash 3× for 3 min at 1000 × $g$ with 0.1% Tx/TBS, plus protease inhibitors. After the final wash, the WGA beads (Vector Laboratories, AL-1023) were eluted with Laemmli Sample Buffer (LSB) at 600 µL per gram of starting material at 99°C for 10 min. The final concentration was 1.11 mg skm/µL beads and LSB. Samples were loaded (beads and LSB) in a 3–15% gradient gel. The proteins were transferred to PVDF-FL membranes (Millipore) as previously published (*Michele et al., 2002*; *Goddeeris et al., 2013*). EDTA (10 mM) was used in the homogenization to extract α-DG containing matriglycan more efficiently in the muscle homogenates (*WT and POMK*), while EDTA had no effect on *Large1myd* α-DG (matriglycan-negative) extraction ( *Large1myd*).

## Immunoblotting and ligand overlay

The mouse monoclonal antibody against matriglycan on α-DG (IIH6, Developmental Studies Hybridoma Bank, University of Iowa; RRID:AB_2617216) was characterized previously and used at 1:100 (*Ervasti and Campbell, 1993*). The polyclonal antibody, AF6868 (R&D Systems, Minneapolis, MN, USA; RRID:AB_10891298), was used at a concentration of 1:100 for immunoblotting the core α-DG and β-DG proteins, and the secondary was a donkey anti-sheep (LI-COR Bioscience, Lincoln, NE, USA) used at 1:10,000 concentration. The antibody against matriglycan on α-DG (III HII) was previously used (*Groh et al., 2009*). Blots were developed with infrared (IR) dye-conjugated secondary antibodies (*Walimbe et al., 2020*) and scanned using the Odyssey infrared imaging system (LI-COR Bioscience). Blot images were captured using the included Odyssey image-analysis software.

Laminin overlay assays were performed as previously described (*Michele et al., 2002*; *Goddeeris et al., 2013*). Immobilon-FL membranes were blocked in laminin-binding buffer (LBB: 10 mM triethanolamine, 140 mM NaCl, 1 mM MgCl$_2$, 1 mM CaCl$_2$, pH 7.6) containing 5% milk followed by incubation with mouse Engelbreth-Holm-Swarm laminin (Thermo Fisher, 23017015) overnight at a concentration of 7.5 nM at 4°C in LBB containing 3% bovine serum albumin (BSA) and 2 mM CaCl$_2$. Membranes were washed and incubated with anti-laminin antibody (L9393; Sigma-Aldrich 1:1000 dilution) followed by IRDye 800 CW dye-conjugated donkey anti-rabbit IgG (LI-COR, 926-32213) at 1:10,000.

## Digestion of α-DG with exoglycosidases

β-Glucuronidase from *T. maritima* and α-xylosidase from *S. solfataricus* were cloned into pET-28a (+) vector between *NheI/XhoI* sites in frame with the N-terminal 6xHis tag. The plasmids (20 ng each) were chemically transformed into 50 µL BL21DE3 One shot competent cells. One colony each was picked and inoculated in 20 mL LB (with kanamycin 50 µg/mL) overnight at 37°C. The next day, 10 mL of the overnight culture was inoculated into 1 L LB (with kanamycin 50 µg/mL). After reaching 0.6 OD at 600 nm the cultures were induced with 1 mM IPTG and incubated at 16°C overnight. The next day the cells were centrifuged at 5000× *g*, for 10 min at 4°C. Cell pellets were stored at –80°C until ready for purification.

The pellets were dissolved in 20 mL homogenization buffer (50 mM Tris-Cl, 150 mM NaCl, 1% TX-100, and all protease inhibitors) per liter culture. The cells were stored again overnight in 50 mL falcon tubes at –80°C for ice crystal formation. Cells were thawed the next day for purification. Nuclease (Pierce) was added at 1.25 kU and cells were sonicated at power level four-five for four times with 10 s intervals in between at 4°C. Cells were then centrifuged at 15,000× *g* for 20 min at 4°C. The supernatant was heat fractionated at 75°C for 10 min after which it was centrifuged at 15,000× *g* for 30 min at 4°C. Meanwhile, a TALON superflow metal affinity column was prepared by packing 3 mL of resin and equilibrating with wash buffer 1 (50 mM Tris-Cl, 100 mM NaCl, 0.1% TX-100, all PIs). All further purification steps were performed at 4°C. The extract was applied to the column three times, such that each time, the extract was incubated with the column for 15–30 min on gentle rocking platform. All flowthrough was saved. The column was washed three times with wash buffer 1. All washes were saved. The column was next washed with high salt wash buffer (50 mM Tris-Cl, 500 mM NaCl, 0.1% TX-100, all PIs) to remove nonspecific interactions and the high salt wash was saved. Proteins were then eluted with elution buffer (50 mM Tris-Cl, 100 mM NaCl, 0.1% TX-100, and 300 mM Imidazole) in five fractions of 3 mL each. The relevant fractions (elutes 1 and 2) were pooled together, and buffer exchanged with 1XPBS pH 7.4 with 30 kDa concentrators (Amicon). One-hundred µL was loaded on SDS-PAGE from all fractions and washes to visualize with Coomassie.

WGA-enriched glycoproteins (elutes) were buffer exchanged with sodium acetate buffer pH 5.5 using 30 kDa concentrators and heated for 5 min in the presence of 10 mM β-mercaptoethanol at 99°C. All protease inhibitors were added after the mixture cooled down. Fifty µL of each enzyme was added per 500 µL of WGA-enriched and buffer-exchanged glycoproteins. The initial time point was aliquoted as $T_o$ and the rest was incubated at 75°C with 600 rpm shaking for 16 hr.

## AAV vector production and AAV injection

The sequence encoding mouse *Large1* was synthesized (Genscript, Piscataway, NJ, USA) and cloned into the AAV backbone under the transcriptional control of the ubiquitous CMV promoter. The AAV2/9 vector contains the genome of serotype 2 packaged in the capsid from serotype 9 and was selected due to its ability to improve muscle transduction efficiency as well as alter tropism. The vector AAV2/9-CMV-*Large1* was generated by the University of Iowa Viral Vector Core Facility. For adult mice, 100 µL ($4.35 \times 10^{12}$ vg) of the vector solution was administered once intraperitoneally or intravenously via the RO sinus. The sequence encoding mouse *Large1* was synthesized (Genscript, Piscataway, NJ, USA) and cloned into the AAV backbone under the transcriptional control of the muscle-specific *MCK* promoter (gift from Jeff Chamberlain). The vector AAV2/9-MCK-*Large1* was generated by the University of Iowa Viral Vector Core Facility. For adult mice, 100 µL ($2.55 \times 10^{12}$ vg) of the vector solution were administered once intraperitoneally or intravenously via the RO sinus. The sequence encoding mouse α-DG lacking the N-terminal domain (H30-A316) was synthesized (Genscript) and cloned into the AAV backbone under the transcriptional control of the muscle-specific *MCK* promoter. The vector AAV2/9-MCK-*DG-E* was generated by the University of Iowa Viral Vector Core Facility. For adult mice, 100 µL ($6.17 \times 10^{11}$ vg) of the vector solution was administered once intraperitoneally or intravenously via the RO sinus. The sequence encoding mouse α-DG N-terminal domain (α-DGN) was synthesized (Genscript) and cloned into the AAV backbone under the transcriptional control of the ubiquitous CMV promoter. The AAV2/9 vector contains the genome of serotype 2 packaged in the capsid from serotype 9 and was selected due to its ability to improve muscle transduction efficiency as well as alter tropism. The vector AAV2/9-CMV-α-DGN was generated by the University of Iowa Viral Vector Core Facility. For adult mice, 100 µL ($1.7 \times 10^{12}$ vg) of the vector solution was administered once intraperitoneally or intravenously via the RO sinus.

## Solid-phase assay

Solid-phase assays were performed as described previously (*Michele et al., 2002*; *Goddeeris et al., 2013*). Briefly, WGA *N*-acetyl-glucosamine buffer eluates were diluted 1:50 in TBS and coated on polystyrene ELISA microplates (Costar 3590) overnight at 4°C. Plates were washed in LBB and blocked for 2 hr in 3% BSA/LBB at room temperature. The wells were washed with 1% BSA/LBB and incubated for 1 hr with L9393 (1:5000 dilution) in 3% BSA/LBB followed by incubation with horseradish peroxidase-conjugated anti-rabbit IgG (Invitrogen, 1:5000 dilution) in 3% BSA/LBB for 30 min. Plates were developed with *o*-phenylenediamine dihydrochloride and $H_2O_2$, and reactions were stopped with 2N $H_2SO_4$. Absorbance per well was read at 490 nm by a microplate reader.

## Statistics

The included Shimadzu post-run software was used to analyze LARGE1 activity in mouse skeletal muscle, and the percent conversion to the product was recorded. The means of three experimental replicates (biological replicates, where each replicate represents a different pair of tissue culture plates or animals, i.e., control and knockout) were calculated using Microsoft Excel, and the mean percent conversion to product for the WT or control sample (control mouse skeletal muscle or M-α-DGN KO mouse skeletal muscle and *Large1^myd* mouse skeletal muscle, respectively) reaction was set to one. The percent conversion of each experimental reaction was subsequently normalized to that of the control, and statistics on normalized values were performed using GraphPad Prism 8. For analysis of LARGE1 activity in mouse skeletal muscle, Student's t-test was used (two-sided). Differences were considered significant at a p-value less than 0.05. Graph images were also created using GraphPad Prism and the data in the present study are shown as the means ± SD unless otherwise indicated. The number of sampled units, n, upon which we report statistics for in vivo data, is the single mouse (one mouse is n=1).

## Materials and data availability

Mouse lines are available from The Jackson Laboratory (see Key resources table) or will be shared upon request. AAV vectors are available upon request. All data needed to evaluate the conclusions in the paper are present in the paper and/or the Supplementary Materials. Mice or AAV vectors should be requested from the corresponding author (kevin-campbell@uiowa.edu).

## Acknowledgements

We thank Keith Garringer for technical assistance and the University of Iowa Viral Vector Core for generating the adeno-associated viral vector (http://www.medicine.uiowa.edu/vectorcore). The *MCK* promoter was a gift from Jeff Chamberlain (University of Washington, Seattle, WA). We are grateful to Dr Jennifer Barr of the Scientific Editing and Research Communication Core at the University of Iowa Carver College of Medicine for her critical reading of the manuscript. We are also grateful to Amber Mower for her assistance with administrative support and Rachel Poe for her support in figure design. This work was supported in part by a Paul D Wellstone Muscular Dystrophy Specialized Research Center grant (1U54NS053672 to KPC). KPC is an investigator of the Howard Hughes Medical Institute. This work was also supported by the Cardiovascular Institutional Research Fellowship (5T32HL007121-45 to JMH). ASW is a fellow in Child Neurology Residency Program at the Baylor College of Medicine. FS was supported by a Grant-in-Aid for Scientific Research C (19K07981) from the Ministry of Education, Culture, Sports, Science and Technology of Japan.

## Additional information

### Funding

| Funder | Grant reference number | Author |
|--------|------------------------|--------|
| Paul D. Wellstone Muscular Dystrophy Specialized Research Center | 1U54NS053672 | Kevin P Campbell |

| Funder | Grant reference number | Author |
| --- | --- | --- |
| Howard Hughes Medical Institute | | Kevin P Campbell |
| Ministry of Education, Culture, Sports, Science and Technology | 19K07981 | Fumiaki Saito |

The funders had no role in study design, data collection and interpretation, or the decision to submit the work for publication.

### Author contributions

Hidehiko Okuma, Kevin P Campbell, Conceptualization, Formal analysis, Supervision, Funding acquisition, Investigation, Methodology, Writing – original draft, Project administration, Writing – review and editing; Jeffrey M Hord, Conceptualization, Formal analysis, Investigation, Methodology, Writing – original draft, Writing – review and editing; Ishita Chandel, David Venzke, Mary E Anderson, Ameya S Walimbe, Soumya Joseph, Formal analysis, Investigation, Methodology, Writing – review and editing; Zeita Gastel, Methodology; Yuji Hara, Fumiaki Saito, Kiichiro Matsumura, Investigation, Methodology

### Author ORCIDs

Hidehiko Okuma ⓘ http://orcid.org/0000-0002-2749-9855
David Venzke ⓘ http://orcid.org/0000-0001-8180-9562
Kevin P Campbell ⓘ http://orcid.org/0000-0003-2066-5889

### Ethics

Animal experimentation: This study was performed in strict accordance with the recommendations in the Guide for the Care and Use of Laboratory Animals of the National Institutes of Health. All animal experiments were approved by the Institutional Animal Care and Use Committee (IACUC) protocols of the University of Iowa (#0081122).

### Decision letter and Author response

Decision letter https://doi.org/10.7554/eLife.82811.sa1
Author response https://doi.org/10.7554/eLife.82811.sa2

## Additional files

### Supplementary files

• MDAR checklist

### Data availability

All data generated or analysed during this study are included in the manuscript and supporting file; Source Data files have been provided for Figures 2C, 5C, 6B, 7B, Figure 2-figure supplement 1, Figure 2-figure supplement 2, Figure 5-figure supplement 1B, and Figure 6-figure supplement 1.

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

# Appendix 1

**Appendix 1—key resources table**

| Reagent type (species) or resource | Designation | Source or reference | Identifiers | Additional information |
|---|---|---|---|---|
| Genetic reagent (*Mus musculus*) | *Pax7^Cre* C57BL/6J | The Jackson Laboratory, Bar Harbor, ME, USA | JAX:010530, RRID:IMSR_JAX:010530 | *Pax7^tm1(cre)Mrc* |
| Genetic reagent (*Mus musculus*) | *Large^myd* | The Jackson Laboratory, Bar Harbor, ME, USA | JAX:000300, RRID:IMSR_JAX:000300 | MYD/Le-Os+/+Large^myd/J |
| Genetic reagent (*Mus musculus*) | *Large^myd* | Campbell Lab | | Described in *Materials and Methods: Animals* (92.5% C57BL/6J) |
| Genetic reagent (*Mus musculus*) | *Dag1^ΔH30-A316* | PMID:31097590 DOI:10.1073/pnas.1904493116 | | *Dag1^Δα-DGN* |
| Genetic reagent (*Mus musculus*) | *Dag1^flox* | PMID:12230980 DOI: 10.1016/s0092-8674(02)00907–8 | JAX:006834, RRID:IMSR_JAX:006834 | B6.129-*Dag1^tm2Kcam*/J |
| Genetic reagent (*Mus musculus*) | *Dag1^T190M* | PMID: 21388311 DOI: 10.1056/NEJMoa1006939 | | |
| Genetic reagent (*Mus musculus*) | α-DGN Tg | This paper | | Described in *Materials and methods: Animals* |
| Genetic reagent (*Mus musculus*) | *Pomk^flox* | PMID:32975514 DOI:10.7554/eLife.61388 | | |
| Antibody | Anti-DG; sheep polyclonal | R&D Systems | Cat# AF6868, RRID:AB_10891298 | WB (1:500) |
| Antibody | Anti-α-DG (IIH6C4); mouse monoclonal | Development Studies Hybridoma Bank/ Campbell Lab | Cat# IIH6 C4, RRID:AB_2617216 | Described in *Materials and Methods: Animals* IF (1:10-1:100) |
| Antibody | Anti-α-DG (IIH6C4); mouse monoclonal | MilliporeSigma Campbell Lab | Cat# 05–593, RRID:AB_309828 | Described in *Materials and Methods: Animals* WB (1:1000–1:2000) |
| Antibody | Anti-Laminin; rabbit polyclonal | MilliporeSigma | Cat# L9393, RRID:AB_477163 | WB (1:1000), Solid Phase Assay (1:5000) |
| Antibody | Anti-β-DG; rabbit polyclonal | Campbell Lab PMID: 1741056 DOI: 10.1038/355696a0 | AP83 | Described in *Materials and Methods: Animals* IF (1:50) |
| Antibody | Anti-β-DG mouse IgM; mouse monoclonal | Leica Biosystems | Cat# NCL-b-DG, RRID:AB_442043 | IF (1:50 to 1:200) |
| Antibody | Anti-sheep IgG; donkey polyclonal | Rockland | Cat# 613-731-168, RRID:AB_220181 | WB (1:2000) |
| Antibody | Anti-mouse IgG (H+L); donkey polyclonal | LI-COR Biosciences | Cat# 926–32212, RRID:AB_621847 | WB (1:15,000), IF (1:800) |

*Appendix 1 Continued on next page*

*Appendix 1 Continued*

| Reagent type (species) or resource | Designation | Source or reference | Identifiers | Additional information |
|---|---|---|---|---|
| Antibody | Anti-rabbit IgG (H+L); donkey polyclonal | LI-COR Biosciences | Cat# 926–32213, RRID:AB_621848 | WB (1:15,000), IF (1:800) |
| Antibody | Anti-mouse IgM; goat polyclonal | LI-COR Biosciences | Cat# 926–32280, RRID:AB_2814919 | WB (1:2500) |
| Antibody | Anti-mouse IgG1; goat polyclonal | LI-COR Biosciences | Cat# 926–32350, RRID:AB_2782997 | WB (1:2000, 1:10,000) |
| Antibody | Anti-rabbit IgG (H+L); goat polyclonal | Thermo Fisher Scientific | Cat# A-11034, RRID:AB_2576217 | IF (1:1000 to 1:2000) |
| Antibody | Anti-mouse IgM; goat polyclonal | Thermo Fisher Scientific | Cat# A-21042, RRID:AB_2535711 | IF (1:1000 to 1:2000) |
| Antibody | Anti-human Synaptophysin (SP11); rabbit monoclonal | Thermo Fisher Scientific | Cat# MA5-14532, RRID:AB_10983675 | IF (1:100) |
| Antibody | Neurofilament NF-H; chicken polyclonal | EnCor Biotechnology | Cat# CPCA-NF-H, RRID:AB_2149761 | IF (1:1000) |
| Antibody | Anti-chicken IgY (H+L); goat polyclonal | Thermo Fisher Scientific | Cat# A32759, RRID:AB_2762829 | IF (1:1000) |
| Antibody | Anti-α-DG (IIIH11); mouse monoclonal | Campbell Lab | | Described in *Materials and Methods: Animals* WB (1:100–1:1000) |
| Chemical compound, drug | Pepstatin A | MilliporeSigma | Cat# 516481 | |
| Chemical compound, drug | Calpain Inhibitor I | MilliporeSigma | Cat# A6185 | |
| Chemical compound, drug | Aprotinin from bovine lung | MilliporeSigma | Cat# A1153 | |
| Chemical compound, drug | Leupeptin | MilliporeSigma | Cat# 108975 | |
| Chemical compound, drug | PMSF | MilliporeSigma | Cat# P7626 | |
| Chemical compound, drug | Immobilon-FL PVDF | MilliporeSigma | Cat# IPFL00010 | |
| Chemical compound, drug | Calpeptin | Thermo Fisher Scientific | Cat# 03-340-05125M | |
| Chemical compound, drug | Bis-acrylamide solution-30% (37.5:1) | Hoefer, Inc | Cat# GR337-500 | |

*Appendix 1 Continued on next page*

*Appendix 1 Continued*

| Reagent type (species) or resource | Designation | Source or reference | Identifiers | Additional information |
|---|---|---|---|---|
| Chemical compound, drug | Benzamidine Hydrochloride Hydrate | MP Biomedicals | Cat# 195068 | |
| Chemical compound, drug | WGA agarose bound | Vector Labs | Cat# AL-1023, RRID:AB_2336862 | |
| Chemical compound, drug | Precision Plus Protein All Blue Standards-500 µL | Bio-Rad | Cat# 161–0373 | |
| Chemical compound, drug | Ethylenediamine Tetraacetic acid, disodium salt dihydrate, EDTA | Thermo Fisher Scientific | Cat# S311-500 | |
| Peptide, recombinant protein | Enzymes, β-glucuronidase α-xylosidase | This paper and PMID: 27526028 DOI: 10.1038/nchembio.2146 | | Described in *Materials and methods: Digestion of α-DG with exoglycosidases* |
| Software, algorithm | SigmaPlot | SigmaPlot | RRID:SCR_003210 | |
| Software, algorithm | Excel | Microsoft | RRID:SCR_016137 | |
| Software, algorithm | GraphPad Prism | GraphPad | RRID:SCR_002798 | Version 8.3 |
| Software, algorithm | FlowJo | Becton, Dickinson & Company (BD) | RRID:SCR_008520 | Version 7.6.5 |
| Software, algorithm | Image Studio Acquisition Software | LI-COR Biosciences | RRID:SCR_015795 | |
| Software, algorithm | Fiji | National Institutes of Health | RRID:SCR_002285 | |
| Software | Adobe Illustrator | Adobe | RRID:SCR_010279 | Version 27.1.1 |
| Software, algorithm | UniProt Proteomes | | RRID:SCR_018666 | |
| Software, algorithm | IUPRED | | RRID:SCR_014632 | |
| Other | Streptavidin, Alexa Fluor 594 conjugate | Thermo Fisher Scientific | Cat# S11227 | IF (1:1000 to 1:2000) |
| Other | Western Blot Imager | LI-COR Biosciences | Odyssey CLx RRID:SCR_014579 | |
| Other | Isolated Mouse Muscle System | Aurora Scientific | 1200A | |
| Other | Mouse Grip Strength Meter | Columbus Instruments | 1027 Mouse | |
| Other | Tabletop ultracentrifuge | Beckman Coulter | Optima MAX, 130K | |
| Other | Ultracentrifuge | Beckman Coulter | Optima-L-100 XP | |
| Other | Centrifuge | Beckman Coulter | Avanti J-E HPC | |

*Appendix 1 Continued on next page*

*Appendix 1 Continued*

| Reagent type (species) or resource | Designation | Source or reference | Identifiers | Additional information |
|---|---|---|---|---|
| Other | Slide Scanner Microscope | Olympus | VS120-S5-FL RRID:SCR_018411 | |
| Other | Confocal Microscope | Olympus | FLUOVIEW FV3000 RRID:SCR_017015 | |
| Other | Cryostat | Leica Biosystems | CM3050S RRID:SCR_016844 | |
| Other | Imaging System | LI-COR Biosciences | Odyssey CLx Infrared RRID:SCR_014579 | |
| Other | 96-Well Plates | Corning Inc | Cat# 3590 | Described in *Materials and methods* |
| Other | Hematoxylin (Certified Biological Stain) | Fisher Scientific | Cat# H345-100 | Described in *Materials and methods* |
| Other | Eosin 515 LT | Leica Biosystems | Cat# 3801619 | Described in *Materials and methods* |
| Other | AAV-MCK DG (ΔH30-A316) | This paper | | Described in *Materials and methods: AAV vector production and AAV injection* |
| Other | AAV-CMV-α-DGN | This paper | | Described in *Materials and methods: AAV vector production and AAV injection* |
| Other | AAV-MCK-*mLarge1* | This paper | | Described in *Materials and methods: AAV vector production and AAV injection* |
| Other | α BTX-488 (α-Bungarotoxin) | Thermo Fisher Scientific | Cat# B13423 | IF (1:500) |
| Other | Laminin (Natural, mouse) Lam-111 | Gibco | Cat# 23017–015 | |
| Other | Enzymes, β-glucuronidase α-xylosidase | This paper and PMID: 27526028 DOI: 10.1038/nchembio.2146 | | Described in *Materials and methods: Digestion of α-DG with exoglycosidases* |

