## [Editor Report]

This study presents fundamental new insight into the process of post-translational modification of α dystroglycan by the protein Large on its N-terminal domain which is critical for concentric muscle contraction. The convincing data presented advances the field beyond a role for POMK in mediating the effect of Large on α dystroglycan, to show that α dystroglycan N-term domain, like POMK itself, is required for LARGE1 to extend matriglycan to its full mature length.

---

## [Decision Letter]

**Decision letter after peer review:**

Thank you for submitting your article "Dystroglycan N-terminal domain enables LARGE1 to extend matriglycan on α-dystroglycan and prevents muscular dystrophy" for consideration by *eLife*. Your article has been reviewed by 2 peer reviewers, and the evaluation has been overseen by a Reviewing Editor and Suzanne Pfeffer as the Senior Editor. The reviewers have opted to remain anonymous.

The reviewers have discussed their reviews with one another, and the Reviewing Editor has drafted this to help you prepare a revised submission. Please address these comments in a revised manuscript, either with direct experimentation or modifications to the text.

Essential revisions:

The reviewers judged that the work would be improved with further clarification of the nature of the interaction between aDGN and LARGE1. If this cannot be tested directly, in the revision then at least the point should be mentioned in the discussion. One of the possible interpretations is that aDGN interacts with LARGE1 for matriglycan synthesis. Given the availability of recent structural information on LARGE1 domains, including one crystal structure in BioRxiv, has any molecular docking with the available structures (see PDB: 1U2C for example, or 4WIQ reporting the structure of the T190M mutant) of the aDGN domain been attempted?

*Reviewer #1 (Recommendations for the authors):*

One important and initial overall comment: this work indeed suggests that LARGE1 can be recognized by aDGN, but could also other sections of the dystroglycan core protein act as LARGE1 chaperones, for example, the mucin-like central region on itself, where the Thr residues on which the M3 glycoepitope is built? In fact, the matriglycan can still be partially elongated or modified when the N-terminal domain is fully depleted. The absence of aDGN must be thus certainly detrimental to the overall affinity displayed by α-DG towards LARGE1, but it does not seem to be a "conditio sine qua non" of glycosylation, for at least some LARGE1 enzymatic activity towards α-dystroglycan does still take place (as stated by the Authors in the Discussion). In this respect, it cannot be ruled out that aDGN, rather than binding LARGE1 directly (see the next point) could be important for the folding and stability of a downstream subdomain/sequence of α-dystroglycan representing the real recognition site for LARGE1. A possible interpretation that the Authors have not mentioned.

As previously discussed, one of the possible interpretations of data is that aDGN would need to interact at some level with LARGE1 for the synthesis of matriglycan. Given the availability of recent structural information on LARGE1 domains, including one crystal structure (see the papers recently deposited as preprints in BioRxiv, one actually deposited by some of the Authors of the paper under review here), has any molecular docking with the available structures (see PDB: 1U2C for example, or 4WIQ reporting the structure of the T190M mutant) of the aDGN domain been attempted?

Figure 2: can the exact number of mice analyzed be reported? Was any difference due to the gender of M-α-DGN KO mice observed? If possible, it would be interesting to report in the figure the gender of the analyzed mice. Lines 115-116: I would suggest amending "M-α-DGN KO mice were lower in weight than WT littermate" to "M-α-DGN KO mice were slightly lower in weight than WT littermate", at least at 12 weeks of age.

Overlay assays (Figure 2C – right panel). Murine laminin-1 was used at 7.5 nM, maybe some lower concentrations could be tried to check whether "shorter" matriglycan might show a reduced affinity for laminin?

Mice aged 12 weeks were analyzed in Figure 2 (in Figure 4 and Figure 5 as well, while in Figure 6, 17-week old and figure 7, 12 to 19 weeks old). Were the mice checked at an older age? Is the observed muscular dystrophy phenotype actually getting worst with age? Histological and immunoblotting data in Figure 2 are reported for quadriceps muscle: was any other muscle analyzed? In Figure 3 (NMJs analysis), for example, tibialis anterior (TA), extensor digitorum longus (EDL), and soleus (SOL) muscles were analyzed in 35-39 weeks old mice, therefore it is reasonable to assume that histology and immunoblot data for the 'old' mice would be available as well.

Figure 5. I agree that it is very interesting to learn that the AAV-driven DGE (lacking aDGN) is matured and shortened similarly to what was observed in M-Dag1-KO muscle. It might sound like an obvious point/suggestion and further control, but is AAV-wt DG able to rescue the phenotypes observed in M-Dag1-KO mice muscle? Figure 5, supplement-1: in the bottom panel of A (M-Dag1-KO), is β-DG staining missing?

Figure 6. Very interesting experiment. The data show that overexpression of isolated aDGN cannot prevent muscular dystrophy in M-α-DGN KO mice. Therefore, I would suggest the authors slightly amend the title of their work because it can lead to the wrong suggestion that aDGN per se can prevent muscular dystrophy. In addition, was aDGN overexpressed in wt animals as well? I think that this would be an important control experiment that must be added, not least because one of the outcomes could be that isolated aDGN interferes with the whole elongation process and somehow "compete" for LARGE1 recognition. Shall we in case expect this to be detrimental to the resulting matriglycan length? In such a scenario, aDGN (the isolated domain) would be a factor inducing muscular dystrophy rather than preventing it.

Figure 6 data again. A technical and important question: how can one be sure that aDGN is overexpressed, and overrepresented, within the Golgi of cells?

The Data in Figure 7 are really interesting because they help establish a direct relationship between the length of matriglycan and the observed dystrophic phenotype. From what is reported, also in Figure 2, it seems that matriglycan reaches some degree of extension in the aDGN KO mice. However, the aDGN KO mice seem to be much more similar to the myd mice in Figure 7, suggesting the idea that under a certain "threshold length" matriglycan would be largely ineffective (i.e. it would not be sufficient for binding). Based also on previously available data on the myd phenotype, can one say that the aDGN KO mice are phenotypically similar to the myd mice or not?

Related to the previous point. In Figure 7, while they are reported in panels D and E, in panels A, B and C no data are reported for myd mice. Can the figure be further updated with these myd mice data? Of course, the myd mice analysed should be of similar age to aDGN KO ones (the age for the analyzed myd mice is not reported, it seems). Panel E is reported in the Figure 7 caption as "I".

One interesting (I believe) question/point of discussion is: are similar binding mechanisms taking place with other dystroglycan modifying enzymes? Indeed, some of the enzymes that are initially involved in the fabrication of the M3 glycoepitope reside in the ER, and they might also need to interact and be correctly positioned towards the DG core protein/mucin-like region for the enzymatic steps they catalyze to be possible. The evidence here reported points instead to aDGN being a "specific chaperone" only for the synthesis of the matriglycan, thus implying that the previous steps in M3 glycoepitope fabrication would be aDGN-independent. Can the Authors discuss this point?

Another interesting question that could become a matter for discussion is "how is aDGN processed/scavenged or transferred to the extracellular space and body fluids (including plasma)?" Can the Authors speculate, or refer to their or perhaps also previously available data, on the presence of a specific protein transfer system for removal of aDGN from the Golgi and then off the cells? Is some internal proteasome-dependent process involved? Or perhaps, the aDGN is processed (by furin) in the Golgi but only removed after α-dystroglycan is transferred to the plasma membrane? In any case, the presence of very limited amounts of aDGN in the plasma seems to suggest that not all the aDGN produced in the cell would be processed through internal degrading pathways, which in itself should be ground for discussion.

*Reviewer #2 (Recommendations for the authors):*

1) The authors may consider exogenously expressing full-length DG (containing α-DGN) n M-Dag1 KO mice to see if matriglycan length and functions are restored. This may strengthen the conclusion that LARGE1 requires α-DGN to synthesize full-length matriglycan.

2) It is not clear why neuromuscular junction immunohistochemistry in Figure 3 was done in markedly older animals (35-39-week-old) animals when all the other assays were carried out in younger animals.

---

## [Author Response]

Essential revisions:The reviewers judged that the work would be improved with further clarification of the nature of the interaction between aDGN and LARGE1. If this cannot be tested directly, in the revision then at least the point should be mentioned in the discussion. One of the possible interpretations is that aDGN interacts with LARGE1 for matriglycan synthesis. Given the availability of recent structural information on LARGE1 domains, including one crystal structure in BioRxiv, has any molecular docking with the available structures (see PDB: 1U2C for example, or 4WIQ reporting the structure of the T190M mutant) of the aDGN domain been attempted?

The reviewers correctly conclude that the interaction between LARGE1 and α-DGN is essential for efficient matriglycan polymerization and that deep-learning tools could be used to predict protein-protein interactions. The authors tried co-folding α-DGN with a LARGE1 dimer only to find that COLABFOLD was not able to form a LARGE1:α-DGN complex. However, our previous *eLife* paper (Walimbe et al., 2020) showed that the LARGE1phosphoglycan interaction is crucial for efficient matriglycan polymerization. Thus, because the interaction between LARGE1 and α-DGN requires post-translational modifications, it is unlikely that deep-learning tools will be able to predict this interaction.

A detailed molecular dissection of the interaction between LARGE1 and α-DGN will require the use of various structural, computational, and biophysical methods; these experiments, including cryo-EM, are presently under progress by another postdoc in the lab and will be described in a future publication from our group.

We have added the following sentences to the discussion to address this comment.

"Thus, the interaction between LARGE1 and α-DGN is essential for efficient matriglycan polymerization. Although deep-learning tools could be used to predict protein-protein interactions, it is unlikely that these tools will be able to predict the interaction between LARGE1 and dystroglycan as this requires posttranslational modifications (Walimbe et al., 2020).”

Reviewer #1 (Recommendations for the authors):One important and initial overall comment: this work indeed suggests that LARGE1 can be recognized by aDGN, but could also other sections of the dystroglycan core protein act as LARGE1 chaperones, for example, the mucin-like central region on itself, where the Thr residues on which the M3 glycoepitope is built? In fact, the matriglycan can still be partially elongated or modified when the N-terminal domain is fully depleted. The absence of aDGN must be thus certainly detrimental to the overall affinity displayed by α-DG towards LARGE1, but it does not seem to be a "conditio sine qua non" of glycosylation, for at least some LARGE1 enzymatic activity towards α-dystroglycan does still take place (as stated by the Authors in the Discussion). In this respect, it cannot be ruled out that aDGN, rather than binding LARGE1 directly (see the next point) could be important for the folding and stability of a downstream subdomain/sequence of α-dystroglycan representing the real recognition site for LARGE1. A possible interpretation that the Authors have not mentioned.

Our previous *eLife* paper (Walimbe et al., 2020) showed that without phosphorylation of core M3 by POMK, LARGE1 synthesizes a very short matriglycan on α-DG. Our current paper shows that a similar short matriglycan is synthesized by LARGE1 in the absence of αDGN. Thus, our working hypothesis is that anchoring LARGE1 to dystroglycan via a core M3 phosphate or α-DGN allows the synthesis of a short matriglycan but that elongation of matriglycan requires both a core M3 phosphate and α-DGN (proof of this hypothesis will require detailed cryo-EM studies).

To address the reviewer’s concern, we have made a transgenic mouse that overexpresses αDGN (α-DGN Tg) and showed that free α-DGN interferes with the elongation of matriglycan, but that LARGE1 can still synthesize a short matriglycan on α-DG (new Figure 5—figure supplement 2). This new data supports our hypothesis and partially addresses the reviewer’s concern.

We have added the following paragraphs and figure to the manuscript to address this comment.

Addition to Methods:

Generation of α-DGN transgenic (α-DGN Tg) mice

“Transgenic mice, which overexpressed only the α-DGN domain of dystroglycan, were generated using C57BL/6NJ mice. First, total RNA was extracted from mouse skeletal muscle to prepare a construct, and RT-PCR obtained the fulllength cDNA of the Dag1 gene containing the Kozak sequence. Using this template, we amplified bases 1-930 of Dag1 corresponding to α-DGN by PCR and inserted a stop codon at the 3' end. The PCR product was cut with restriction enzymes and cloned into an EcoR I / Not I site of plasmid pCAGGS (Unitech). In addition, a fragment containing the CAG promoter and Dag1 1-930 was cut out from the same vector using Sal 1/Hind III and inserted into a vector for microinjection. The vector was microinjected into a pronuclear stage of zygotes of wild-type mice and transferred to pseudopregnant females. Offspring were screened for genomic integration of this fragment by PCR of tail DNA, using the following unique sequence of the vector: forward 5′-CCTACAGCTCCTGGGCAACGTGCTGGTT-3′, reverse 5′-AGAGGGAAAAAGATCTCAGTGGTAT-3′. Mice were generated by breeding F1 heterozygous transgenic males to wild-type females.”

Addition to Results:

“To test if excessive free α-DGN interferes with the binding of endogenous α-DGN to LARGE1, we analyzed immunoblots of skeletal muscle from control mice and those overexpressing free α-DGN (α-DGN Tg). Immunoblot analysis of these mice showed that α-DG had a reduced molecular weight of ~100-125 kDa, whereas β-DG remained unchanged (Figure 5—figure supplement 2). This biochemical phenotype is similar to that observed in the skeletal muscles of M-αDGN KO mice and demonstrates that free α-DGN interferes with the elongation of matriglycan, but LARGE1 can still synthesize the short matriglycan on α-DG.”

As previously discussed, one of the possible interpretations of data is that aDGN would need to interact at some level with LARGE1 for the synthesis of matriglycan. Given the availability of recent structural information on LARGE1 domains, including one crystal structure (see the papers recently deposited as preprints in BioRxiv, one actually deposited by some of the Authors of the paper under review here), has any molecular docking with the available structures (see PDB: 1U2C for example, or 4WIQ reporting the structure of the T190M mutant) of the aDGN domain been attempted?

The reviewers correctly conclude that the interaction between LARGE1 and α-DGN is essential for efficient matriglycan polymerization and that deep-learning tools could be used to predict protein-protein interactions. The authors tried co-folding α-DGN with a LARGE1 dimer only to find that COLABFOLD was not able to form a LARGE1:α-DGN complex. Our previous *eLife* paper (Walimbe et al., 2020) showed that the LARGE1-phosphoglycan interaction is crucial for efficient matriglycan polymerization. Thus, it is unlikely deeplearning tools will be able to predict an interaction between LARGE1 and α-DGN as this requires post-translational modifications.

Another postdoc in the lab is currently conducting a detailed molecular dissection of the interaction between LARGE1 and α-DGN using various structural, computational, and biophysical methods, including cryo-EM, the results of which will appear in a future publication from our group.

We have added the following sentences to the discussion to address this comment.

"Thus, the interaction between LARGE1 and α-DGN is essential for efficient matriglycan polymerization. Although deep-learning tools could be used to predict protein-protein interactions, it is unlikely that these tools will be able to predict the interaction between LARGE1 and dystroglycan as this requires posttranslational modifications (Walimbe et al., 2020).”

Figure 2: can the exact number of mice analyzed be reported? Was any difference due to the gender of M-α-DGN KO mice observed? If possible, it would be interesting to report in the figure the gender of the analyzed mice. Lines 115-116: I would suggest amending "M-α-DGN KO mice were lower in weight than WT littermate" to "M-α-DGN KO mice were slightly lower in weight than WT littermate", at least at 12 weeks of age.

The number of KO mice used in these experiments was 17, and that of LC mice was 57. There were six male KO mice and 11 female KO mice. There were 23 male LC mice and 34 female LC mice.

We have updated our manuscript to include the number of mice analyzed, as indicated above. We have also added “slightly” in the description of the results, as suggested.

Overlay assays (Figure 2C – right panel). Murine laminin-1 was used at 7.5 nM, maybe some lower concentrations could be tried to check whether "shorter" matriglycan might show a reduced affinity for laminin?

Previously, we have shown a direct correlation between matriglycan length and its binding capacity and affinity for extracellular matrix ligands (Goddeeris et al., 2013). We have not tried lower concentrations of laminin in the overlay assay, but we did use lower concentrations of laminin in the solid phase binding assay (see Figure 7D and E) and calculated relative Bmax values (Figure 7D) and K_d_ values (Figure 7D and figure legend). These results were also mentioned in the text.

Mice aged 12 weeks were analyzed in Figure 2 (in Figure 4 and Figure 5 as well, while in Figure 6, 17-week old and figure 7, 12 to 19 weeks old). Were the mice checked at an older age? Is the observed muscular dystrophy phenotype actually getting worst with age? Histological and immunoblotting data in Figure 2 are reported for quadriceps muscle: was any other muscle analyzed? In Figure 3 (NMJs analysis), for example, tibialis anterior (TA), extensor digitorum longus (EDL), and soleus (SOL) muscles were analyzed in 35-39 weeks old mice, therefore it is reasonable to assume that histology and immunoblot data for the 'old' mice would be available as well.

We have not done a detailed natural history study, but the muscular dystrophy phenotype does get more severe with age. In addition to quadriceps muscle, we have analyzed diaphragm, iliopsoas, and EDL as well. We chose to show the data from the quadriceps muscle because it had a more severe phenotype than other muscles.

Figure 5. I agree that it is very interesting to learn that the AAV-driven DGE (lacking aDGN) is matured and shortened similarly to what was observed in M-Dag1-KO muscle. It might sound like an obvious point/suggestion and further control, but is AAV-wt DG able to rescue the phenotypes observed in M-Dag1-KO mice muscle? Figure 5, supplement-1: in the bottom panel of A (M-Dag1-KO), is β-DG staining missing?

Our previous studies (Hara et al., 2011, PNAS; Hara et al., 2011, NEJM) showed that adenovirus expression of WT-DG in DAG1 knockout muscle cells or mice was able to rescue the phenotype. In Figure 5—figure supplement 1, the muscle of M-Dag1-KO showed no β-DG staining because DAG1 (α-DG and β-DG) was removed.

Figure 6. Very interesting experiment. The data show that overexpression of isolated aDGN cannot prevent muscular dystrophy in M-α-DGN KO mice. Therefore, I would suggest the authors slightly amend the title of their work because it can lead to the wrong suggestion that aDGN per se can prevent muscular dystrophy. In addition, was aDGN overexpressed in wt animals as well? I think that this would be an important control experiment that must be added, not least because one of the outcomes could be that isolated aDGN interferes with the whole elongation process and somehow "compete" for LARGE1 recognition. Shall we in case expect this to be detrimental to the resulting matriglycan length? In such a scenario, aDGN (the isolated domain) would be a factor inducing muscular dystrophy rather than preventing it.

We have changed the title of the manuscript to “N-terminal domain on dystroglycan enables LARGE1 to extend matriglycan on α-dystroglycan and prevents muscular dystrophy” as suggested.

To address the reviewer’s concern, we have made a transgenic mouse that overexpresses αDGN (α-DGN Tg) and showed that free α-DGN interferes with the elongation of matriglycan, but that LARGE1 can still synthesize a short matriglycan on α-DG (new Figure 5—figure supplement 2). This new data supports our hypothesis and partially addresses the reviewer’s concern.

Please see the addition to the manuscript (Methods, Results, and new Figure 5—figure supplement 2) in response to Reviewer #1’s first comment.

Figure 6 data again. A technical and important question: how can one be sure that aDGN is overexpressed, and overrepresented, within the Golgi of cells?

Our antibody to α-DGN has not worked in IF studies so we cannot demonstrate Golgi localization.

The Data in Figure 7 are really interesting because they help establish a direct relationship between the length of matriglycan and the observed dystrophic phenotype. From what is reported, also in Figure 2, it seems that matriglycan reaches some degree of extension in the aDGN KO mice. However, the aDGN KO mice seem to be much more similar to the myd mice in Figure 7, suggesting the idea that under a certain "threshold length" matriglycan would be largely ineffective (i.e. it would not be sufficient for binding). Based also on previously available data on the myd phenotype, can one say that the aDGN KO mice are phenotypically similar to the myd mice or not?

The phenotype of the α-DGN KO mice reflects that of POMK KO mice in that they have matriglycan of similar lengths, as shown in the paper. The short matriglycan may be able to maintain the linkage of the cytoskeleton to the ECM, thus differentiating the phenotype from myd mice (LARGE1 knockout mice), which do not have matriglycan.

Related to the previous point. In Figure 7, while they are reported in panels D and E, in panels A, B and C no data are reported for myd mice. Can the figure be further updated with these myd mice data? Of course, the myd mice analysed should be of similar age to aDGN KO ones (the age for the analyzed myd mice is not reported, it seems). Panel E is reported in the Figure 7 caption as "I".

Data on myd mice have been published in many papers (Michele et al., 2002; Goddeeris et al., 2013; Walimbe et al., 2020) so we have not included these data in all our figures.

We thank the reviewer for pointing out the discrepancy between panel labels and the figure legend and have changed the caption to refer to the correct panel (e.g., changed from “I” to ”E”).

One interesting (I believe) question/point of discussion is: are similar binding mechanisms taking place with other dystroglycan modifying enzymes? Indeed, some of the enzymes that are initially involved in the fabrication of the M3 glycoepitope reside in the ER, and they might also need to interact and be correctly positioned towards the DG core protein/mucin-like region for the enzymatic steps they catalyze to be possible. The evidence here reported points instead to aDGN being a "specific chaperone" only for the synthesis of the matriglycan, thus implying that the previous steps in M3 glycoepitope fabrication would be aDGN-independent. Can the Authors discuss this point?

Matriglycan-producing enzymes go through a stepwise sequence, with the final step being that LARGE1 elongates matriglycan on αDG. Therefore, we believe αDGN affects only LARGE1 and is involved in the elongation of matriglycan.

Another interesting question that could become a matter for discussion is "how is aDGN processed/scavenged or transferred to the extracellular space and body fluids (including plasma)?" Can the Authors speculate, or refer to their or perhaps also previously available data, on the presence of a specific protein transfer system for removal of aDGN from the Golgi and then off the cells? Is some internal proteasome-dependent process involved? Or perhaps, the aDGN is processed (by furin) in the Golgi but only removed after α-dystroglycan is transferred to the plasma membrane? In any case, the presence of very limited amounts of aDGN in the plasma seems to suggest that not all the aDGN produced in the cell would be processed through internal degrading pathways, which in itself should be ground for discussion.

We have added the following sentences to the discussion to address this comment.

“α-DGN is cleaved by the proprotein convertase furin at the sequence RVRR (amino acids 309–312) (Singh et al., 2004). It has a globular structure that is organized into two subdomains (Brancaccio et al., 1997). The first subdomain is a typical Ig-like domain; the second subdomain resembles ribosomal RNA-binding proteins (Bozic et al., 2004). α-DGN is secreted by cells in culture (Saito et al., 2011) and has been detected in a wide variety of human bodily fluids, including serum and plasma (Saito et al., 2008; Saito et al., 2011). Decreased levels of αDGN have been reported in the serum of patients with Duchenne muscular dystrophy (DMD) and in the serum of utrophin-deficient mdx mice, a mouse model for DMD (Crowe et al., 2016). Finally, we have also studied the protective role of secreted α-DGN (de Greef et al., 2019) on influenza A virus proliferation.”

Reviewer #2 (Recommendations for the authors):1) The authors may consider exogenously expressing full-length DG (containing α-DGN) n M-Dag1 KO mice to see if matriglycan length and functions are restored. This may strengthen the conclusion that LARGE1 requires α-DGN to synthesize full-length matriglycan.

Our previous studies (Hara et al., 2011, PNAS; Hara et al., 2011, NEJM) showed that adenovirus expression of WT-DG in DAG1 knockout muscle cells or mice was able to rescue the phenotype. We agree that this supports our conclusion that LARGE1 requires α-DGN to synthesize full-length matriglycan.

We have added the following sentences to the discussion to address this comment.

“We previously reported that WT dystroglycan is able to restore full-length matriglycan and its function when expressed in DG null cells or muscle.”

2) It is not clear why neuromuscular junction immunohistochemistry in Figure 3 was done in markedly older animals (35-39-week-old) animals when all the other assays were carried out in younger animals.

We studied older mice since the NMJ phenotype was more severe with age and now included this information in the text.